# Global genomic population structure of wild and cultivated oat reveals signatures of chromosome rearrangements

Wubishet A. Bekele [1] ✉, Raz Avni [2,33], Clayton L. Birkett [3,33], Asuka Itaya [1,33], Charlene P. Wight [1,33], Justin Bellavance[1], Sophie Brodführer[4], Francisco J. Canales [5], Craig H. Carlson [6], Anne Fiebig [2], Yongle Li[7], Steve Michel[8], Raja Sekhar Nandety [6], David J. Waring [9], Juan D. Arbelaez[10], Aaron D. Beattie[11], Melanie Caffe[12], Isabel A. del Blanco [13], Jason D. Fiedler [6], Rajeev Gupta [6], Lucia Gutierrez [14], John C. Harris [15], Stephen A. Harrison[16], Matthias H. Herrmann [4], Yung-Fen Huang [17], Julio Isidro y Sanchez [18], Michael S. McMullen[19], Jennifer W. Mitchell Fetch[20], Kirby T. Nilsen[20,21], Isobel A. P. Parkin [22], YuanYing Peng[23], Kevin P. Smith[24], Tim Sutton [15,25], Weikai Yan[1], Pamela Zwer[15], Axel Diederichsen[26,33], Kathy Esvelt Klos[27,33], Yong-Bi Fu [22,33], Catherine J. Howarth [28,33], Jean-Luc Jannink[3,9,33], Eric N. Jellen [29,33], Tim Langdon[28,33], Peter J. Maughan[29,33], Edyta Paczos-Grzeda [30,33], Elena Prats[5,33], Taner Z. Sen[8,31,33], Martin Mascher [2,32] ✉ & Nicholas A. Tinker [1] ✉

The genus *Avena* consists of approximately 30 wild and cultivated oat species. Cultivated oat is an important food crop, yet the broader genetic diversity within the *Avena* gene pool remains underexplored and underexploited. Here, we characterize over 9000 wild and cultivated hexaploid oat accessions of global origin using genotyping-by-sequencing and explore population structure using multidimensional scaling and population-based clustering methods. We also conduct analyses to reveal chromosome regions associated with local adaptation, sometimes resulting from large-scale chromosome rearrangements. We report four distinct genetic populations within the wild species *A. sterilis*, a distinct population of cultivated *A. byzantina*, and multiple populations within cultivated *A. sativa*. Some chromosome regions associated with local adaptation are also associated with confirmed structural rearrangements on chromosomes 1A, 1C, 3C, 4C, and 7D. This work provides evidence suggesting multiple polyploid origins, multiple domestications, and/or reproductive barriers amongst *Avena* populations caused by differential chromosome structure.

The oat genus *Avena*, with a basic chromosome number of seven, includes diploid, tetraploid, and hexaploid species. These species have haploid genome types classified as A, B, C, or D[1]. Taxonomic classification in *Avena* has been controversial, but a recent review settles on 30 *Avena* species[2], of which six are hexaploids having a haploid genome constitution of ACD. These include *A. sterilis* L., *A. sativa* L., *A. byzantina* Koch, *A. fatua* L., *A. occidentalis* Dur., and *A. ludoviciana* Dur. *Avena byzantina* has occasionally been considered as a subspecies of *A.*

*sativa*, while additional hexaploid species (e.g., *A. hybrida* Peterm.) have been distinguished by some taxonomists[2]. Crosses can be made among all hexaploid species, with varying degrees of hybrid fertility[3]. The ACD genome is most likely derived from non-reduced hybridization(s) of ancestors related to the diploid *A. longiglumis* Durieu (A) and the tetraploid *A. insularis* Ladiz. (CD), but no naturally occurring diploid D-genome species have been identified[4]. The two cultivated hexaploid species, *A. sativa* and *A. byzantina*, were domesticated from the wild species *A. sterilis*, which is found in wild populations throughout the Mediterranean basin, North Africa, and the Middle East[5]. The common weedy species *A. fatua*, as well as *A. occidentalis* and *A. hybrida*, all have a floret shattering character that contrasts with the spikelet shattering character found in *A. sterilis* and *A. ludoviciana*[6].

The genetic and genomic analysis of oat has received widespread attention in recent years due to the recognized contributions of oat to healthy human diets[7] and to scientific interest in polyploid genome evolution[8]. The analysis of genetic diversity in populations using genome-wide DNA markers has become a standard and indispensable research tool. It has been widely applied in many plant species, including wild and cultivated hexaploid oat. Examples of applications include making informed decisions about germplasm improvement[9], quantifying genetic erosion or bottlenecks[10], identifying gaps or duplications in germplasm collections[11], designing and interpreting genome-wide association studies[12], and elucidating evolutionary or domestication events[13].

Previous analyses have shown that modern cultivated oat germplasm that is adapted to specific geographical regions such as Canada[14] and Europe[15] has not fully utilized the diversity that is available in collections of older landraces[16–18]. There is also a distinct separation between germplasm from regions where oat is grown during winter seasons *vs.* germplasm adapted to temperate, spring-planted environments[19]. While no clear genetic separation was observed between covered *vs.* hulless oat in North American germplasm[19], Chinese hulless oat landraces have been identified as a distinct category with limited genetic diversity[20]. Interestingly, a recent study suggested that Chinese hulless landraces occupy a much greater diversity space than does covered oat[21]. However, the latter inference was based on only 22 globally sourced covered lines. Two studies have reported a joint analysis of white (*A. sativa*) and red (*A. byzantina*) oat[22,23], with both reporting these to be genetically distinct types, possibly justifying the renewed use of two separate species names.

A limited number of studies have addressed genetic diversity in *A. sterilis*. Rezai and Frey[24] observed striking phenotypic differences among *A. sterilis* populations. For example, the species could be divided into eastern *vs.* western Mediterranean lines based on the number of spikelets per panicle. Goffreda et al.[25] identified a large genetic divergence of accessions from Iran and Iraq relative to accessions from more western regions around the Mediterranean Sea. However, a subsequent study found that patterns of diversity were generally not congruent with geographical origin[26]. In contrast, Volis et al.[27] reported a reduction in diversity and an increase in differentiated alleles as one moves from the species core in the western Mediterranean eastward to the edge of natural *A. sterilis* habitats in Israel. This partially supports a study of Jordanian *A. sterilis*[28] that reported three distinct clusters of diversity within this relatively small country. Overall, genetic studies of *A. sterilis* remain inconclusive and would benefit from larger samples, higher marker density, and a global comparison to cultivated red and white oat.

Arguably, all previous genomic diversity studies in oat have been limited by one or more factors including: sample size, sample diversity, marker density, or lack of a reference genome. Marker density has increased with the development of recent SNP platforms, including array-based genotyping platforms[29] and genotyping by sequencing (GBS)[30]. Because of its lower cost, GBS has been the SNP platform of choice in many studies. We now have the added advantage of having fully annotated hexaploid oat reference genomes[8,31], as well as a pan-genome analysis based on 33 *Avena* accessions, reported in a companion paper[32].

In this work, we assemble and analyze a GBS data set that captures the genetic diversity in a large and globally representative set of wild and cultivated hexaploid oat and guides the selection of a representative pangenome. We identify population structure and its possible associations with speciation, adaptation, and chromosome structure. We investigate the diversity and uniqueness of regional oat improvement programs relative to global oat diversity. We support further investigations through accessible public data repositories and web-enabled visualization tools. Our results show four distinct genetic populations within the wild species *A. sterilis*, a distinct population of cultivated *A. byzantina*, and multiple populations within cultivated *A. sativa*. We find that some chromosome regions associated with local adaptation are also associated with confirmed structural rearrangements on chromosomes 1A, 1C, 3C, 4C, and 7D. This work provides evidence suggesting multiple polyploid origins, multiple domestications, and/or reproductive barriers amongst *Avena* populations caused by differential chromosome structure.

## Results
### Data summary
Metadata for 9,153 diverse hexaploid oat taxa from 15 experiments (Table 1, Supplementary Table 1) were curated on a best-effort basis (Supplementary Data 1). The metadata also includes sequence alignment statistics, completeness of genotypes, and whether each taxon was included in each of three data sets. Where possible, these data include collection sites for non-cultivated oat accessions and landraces, or breeder location for cultivated *A. sativa* accessions. Species names assigned by the genebank or the original collector were preserved (see Supplementary Note 1). However, we expect updates or corrections will be made to the passport data as readers engage with this work. These will be added as marked revisions to the supplements on a project page (https://graingenes.org/GG3/content/global-oat-genomic-diversity-project) on the GrainGenes database[33].

The primary filtered GBS SNP calls (Matrix50) contained 9112 taxa and 115,482 sites. The filtered data set with imputed genotypes contained 8816 taxa and 19,928 sites and was used for most downstream analyses. All accessions contained SNPs on all 21 chromosomes, confirming the hexaploid nature of all samples. The SNP identifiers from the 'Sang' genome were cross-referenced with reference-free Haplotag SNP identifiers[34] because the latter have been widely used in previous publications (Supplementary Data 2).

Duplicated accessions are a common and complex issue in genebanks and biodiversity analyses[35]. Our work included 99 intentionally resampled genebank accessions, and 648 potentially duplicated accessions based on identical or similar names. These duplicated accessions provided an opportunity to compare variability within genebank accessions to that of cultivated oat varieties acquired through different seed sources. Further analysis of these duplicated samples is provided as Supplementary Discussion 1, together with Supplementary Data 3 and Supplementary Fig. 1. As discussed, no duplicates were removed from this study.

A phenotypic data analysis was beyond the scope of this study. However, phenotypic data are available for many of the published collections of taxa used in this study (Table 1). We also summarized all available phenotypic data from accessions with records in the oat collection maintained at Plant Gene Resources of Canada, averaging these by the 21 population groups reported later (Supplementary Data 4). Some interesting comparisons among populations can be seen in this summary. For example, domestication traits such as shattering and hull characteristics are clearly different between cultivated and non-cultivated groups, and there are potential differences in juvenile growth habits among different *sterilis* groups. Plant height and several

**Table 1 | Experiments contributing to global analysis of oat genomic diversity**

| Name[a] | Size[b] | Size[c] | Reference[d] | Description |
|---|---|---|---|---|
| Ancestors | 439 | 439 | Yan et al.[4] | Hexaploid accessions from diverse oat landraces and wild relatives along with additional samples from Fu and Tinker (unpublished) |
| Aberystwyth | 150 | 149 | Langdon, Howarth | Mixed experiment: *A. sterilis* and old UK cultivars and landraces |
| Australia | 561 | 545 | Sutton, Harris, and Zwer | Diversity panel of advanced Australian breeding material and cultivars (2 sets) |
| Biomob | 1342 | 1332 | Parkin | Collection from Plant Gene Resources of Canada, AAFC: mostly *A. sterilis* |
| KLAR | 534 | 534 | Brodführer et al.[68] | Collection of European oat cultivars and landraces, mostly from central Europe |
| China | 190 | 190 | Yan et al.[20] | Cultivars and landraces from China |
| CORE | 676 | 669 | Esvelt Klos et al.[19] | Collection of oat cultivars. Includes AFRI (global), spring, and winter lines (identified in Supplementary Data 1) |
| Cornell | 1010 | 1006 | Carlson et al.[69] | Collection of lines from the CORE panel (overlaps) and from Iowa State university |
| IOI | 230 | 229 | Tinker | Diverse oat founder line collection of landraces and cultivars. Some overlap with CORE |
| Jordan | 275 | 0 | Al-Hajaj et al.[28] | Collected *A. sterilis* accessions from Jordan. Approximately 10 lines per collection site |
| Mediterranean | 708 | 708 | Canales et al.[22] | Collection of Spanish landraces and Mediterranean *A. sativa* and *A. byzantina* lines |
| NAM | 18 | 18 | Langdon | Founder lines from a nested association mapping panel |
| Fusarium | 190 | 190 | Isidro-Sánchez et al.[12] | Collection of oat cultivars from Northern and central Europe |
| NSGC | 368 | 355 | Winkler et al.[18] | Subsample from a diverse oat landrace panel (GBS data generated post publication by Huang) |
| POGI | 2477 | 2452 | Jannink, Fiedler, and Tinker | Material from the Public Oat Genotyping Initiative, including adapted breeding lines from multiple North American breeding programs (identified in Supplementary 1) |

[a]Experiments are described further in Supplementary Table 1.
[b]Full size of filtered population in Matrix50 ($N = 9168$). Available metadata are described in Supplementary Data 1.
[c]Number included in the filtered, imputed data set Matrix80 ($N = 8817$).
[d]Reference for population or lead contacts (from author list) for previously unpublished data.

panic characters also differed between population P05 (see next section) and the other *A. sterilis* groups. We encourage the use of this table to develop further hypotheses, but we draw attention to the population sizes ($N$) for which observations are available, and to the non-orthogonality of these data, which may preclude many types of statistical analyses.

Supplementary Data 1 records which taxa were included in each data set, as well as the number of missing genotypes for that taxa within each data set. This table also records the number of aligned reads that contributed to the genotyping of each taxa. In Matrix50, the minimum number of aligned reads was only 37431, while the minimum number of aligned reads for inclusion in Matrix80 was 150127. While some taxa had large amounts of missing genotypes (especially those from the Jordanian experiment, Table 1) and were therefore omitted from Matrix80 and from most further analyses, we still wanted to visualize genetic similarity among all taxa using a factorial analysis. In Supplementary Discussion 2, Supplementary Method 1 and 2, and Supplementary Fig. 2, we compare Principal Components Analysis (PCA) vs. Multidimensional Scaling (MDS) using two alternate methods of SNP calling and two levels of filtering. This analysis showed that MDS was robust and consistent as a descriptive method. Thus, we employed MDS analysis within the full data set for data visualizations. Supplementary Discussion 2 and Supplementary Table 2 also demonstrate that the use of an *A. sativa* reference genome did not adversely affect SNP calling rate in either *A. byzantina* or *A. sterilis*.

Population structure analysis was then performed using the fast and efficient *sNMF* method for $K = 1$ to 40 based on the imputed and filtered Matrix80 data set. The cross-entropy plot (Supplementary Fig. 3) showed no obvious plateau or optimum. Values of $K = 12, 16$, and 21 were selected to explore population structure within the data. A value of $K = 21$ provided good separation of geographic and/or biological clusters (e.g., separating species or breeding programs) and was selected for further data exploration. Other values of $K$ (12 and 16) and an alternate analysis using Discriminant Analysis of Principal Components (DAPC) and $K$-means clustering are included in source data for Supplementary Fig. 3.

## Patterns of global oat genomic diversity

The inferred populations based on sNMF with $K = 21$ were re-numbered, beginning with groups containing predominantly *A. byzantina* and *A. sterilis*. Population P01 contains most of the known *A. byzantina* accessions, forming a distinct group separated from *A. sativa*, while P02 to P05 contain most lines classified as *A. sterilis*. The remaining populations are primarily composed of *A. sativa* accessions (Fig. 1a).

The five populations identified as *A. byzantina* and *A. sterilis*, as well as P06 (composed of *A. sativa* landraces from Spain) are highlighted on a projection of a three-dimensional MDS plot (Fig. 1b). A rotating visualization of these data (https://graingenes.shinyapps.io/Avena_diversity/) allows users to selectively highlight taxa based on species, experiment, origin, or inferred population membership. Here, *A. byzantina* (P01) and one *A. sterilis* population (P02) are effectively separated on the first two axes, while three *A. sterilis* populations (P03, P04, and P05) are separated from other populations but partially overlapping with each other. The *A. sterilis* population P05 also appears to overlap with a population of Spanish *A. sativa* landraces (P06). The admixture plot (Supplementary Fig. 4) suggests that populations P03 and P04 have distinct allele frequencies and are not admixed. Likewise, P05 appears to be the closest *A. sterilis* population to other *A. sativa* populations, while the admixture analysis does not show substantial evidence that P05 is genetically admixed with any other populations (Supplementary Fig. 5).

Figure 1c shows the collection sites of the *A. sterilis* accessions. Interactive exploration of these collection sites can be performed at https://www.google.com/maps/d/u/0/edit?mid=1Ocpmx6jzvWq7BwFEL4Y5e_F4QODxIU&usp=sharing. Here, it appears that *A. sterilis* populations P03 and P04 originated in the northern and eastern Mediterranean regions *vs.* northwest Africa, respectively. The geographical origins of population P02 appear similar to those of P03, except that P02 extends further eastward into northern Iran. Population P05 appears to originate more exclusively in regions of Iran, Iraq, Turkey, Armenia, and Georgia. As most *A. sterilis* accessions (1355 of 1727) were analyzed in a single GBS experiment, the possibility of experimental bias could largely be ruled out (see

**a**

| Species | P01 | P02 | P03 | P04 | P05 | P06 | P07 | P08 | P09 | P10 | P11 | P12 | P13 | P14 | P15 | P16 | P17 | P18 | P19 | P20 | P21 | Sum |
|---|---|---|---|---|---|---|---|---|---|---|---|---|---|---|---|---|---|---|---|---|---|---|
| *A. byzantina* | 99 | 0 | 0 | 0 | 0 | 0 | 0 | 1 | 0 | 0 | 0 | 0 | 0 | 0 | 0 | 0 | 0 | 0 | 0 | 0 | 0 | 354 |
| *A. sterilis* | 0 | 18 | 37 | 35 | 9 | 0 | 0 | 0 | 0 | 0 | 0 | 0 | 0 | 0 | 0 | 0 | 0 | 0 | 0 | 0 | 0 | 1407 |
| *A. sativa* | 0 | 0 | 0 | 0 | 0 | 1 | 8 | 8 | 10 | 1 | 15 | 2 | 3 | 5 | 5 | 10 | 8 | 5 | 6 | 7 | 5 | 6970 |
| *A. fatua* | 0 | 10 | 0 | 8 | 26 | 3 | 0 | 0 | 3 | 0 | 10 | 26 | 0 | 0 | 0 | 0 | 3 | 0 | 0 | 0 | 13 | 39 |
| *A. hybrida* | 0 | 0 | 0 | 0 | 67 | 0 | 0 | 0 | 0 | 5 | 5 | 10 | 0 | 0 | 0 | 0 | 5 | 0 | 5 | 0 | 5 | 21 |
| *A. occidentalis* | 0 | 0 | 0 | 95 | 0 | 0 | 0 | 0 | 0 | 0 | 0 | 5 | 0 | 0 | 0 | 0 | 0 | 0 | 0 | 0 | 0 | 19 |
| *A. ludoviciana* | 0 | 0 | 0 | 0 | 0 | 0 | 0 | 0 | 0 | 0 | 0 | 0 | 0 | 0 | 0 | 0 | 0 | 0 | 0 | 0 | 100 | 2 |

**b**

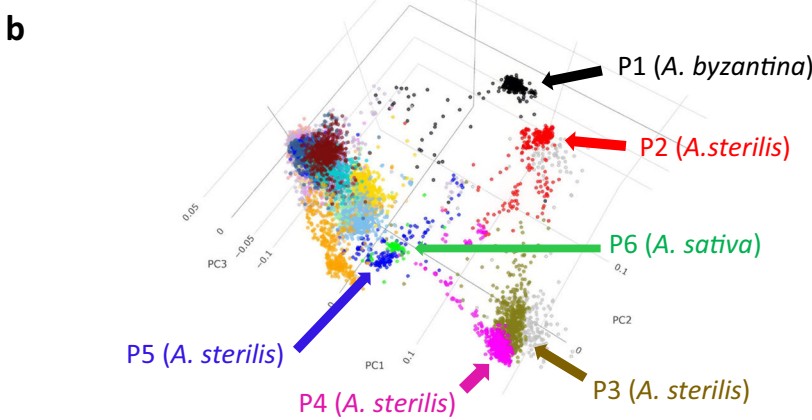

**c**

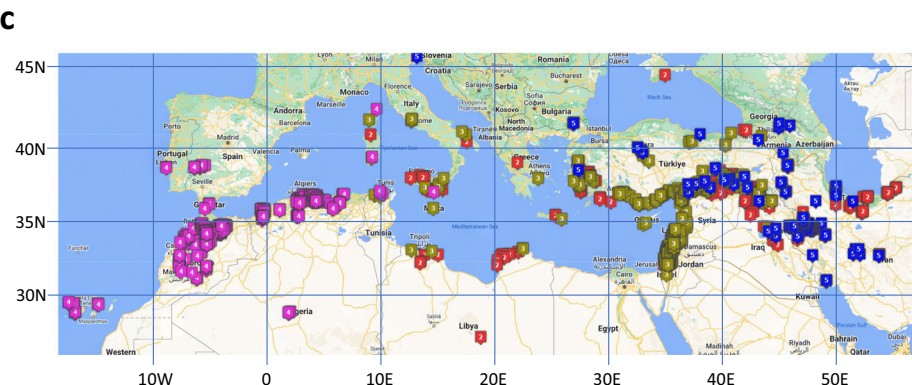

**Fig. 1 | Populations of collected wild oat. a** Heatmap of taxa percentages by species in each of $K = 21$ populations from sparse nonnegative matrix factorization (sNMF) analysis. Light blue indicates a low percentage (2–10%) and dark blue indicates a high percentage (over 50%) while no color indicates 0–1%. The sum is the total number of taxa. Percentages may not sum to 100 due to rounding differences. *A. hausknechtii*, a contested species with a single duplicated accession in P04 is omitted. **b** Multi-dimensional scaling of $n = 9112$ taxa, colored by population membership. Four populations composed primarily of *Avena sterilis* (P02, P03, P04, P05) are indicated, as well as the population of *A. byzantina* (P01) and a population of *A. sativa* landraces from Spain (P06). The remaining non-labeled populations are primarily *A. sativa*. An interactive, rotatable version of this plot is available online (http://graingenes.shinyapps.io/Avena_diversity/). **c** Map of collection sites for taxa from four *A. sterilis* populations. An interactive version of this map is available at https://www.google.com/maps/d/edit?mid=1kYjxF-c5K-VHeIth0oSCwz5zt6-P5Yo&usp=sharing. Source data are provided as a Source Data file.

Supplementary Note 2). Nevertheless, more extensive collections would likely increase the resolution and extent of these regional patterns.

A scatter plot and Mantel test (Supplementary Fig. 6) supported a weak-but-significant positive correlation ($r = 0.1$, $p = 0.001$) between geographic distances *vs.* genetic distances. This correlation is weak because some geographically distant populations of *A. sterilis* (e.g., P03 *vs.* P04) share more genetic similarity than some proximal populations (e.g., P02 *vs.* P05). This raises the question of why different populations (e.g., P02 and P05) maintain distinctness even though they exist in overlapping geographical regions. Although all hexaploids are considered to be interfertile[3], further experiments using designed crosses between populations could be conducted now that we have delineated these populations.

We also examined the population structure and geographic distribution of non-cultivated species with minor representation (Fig. 1a). The 34 *A. fatua* accessions fell into eight different populations, including those previously identified as both *A. sativa* and *A. sterilis*. *A. hybrida* fell mostly into P05, while *A. occidentalis* fell almost exclusively into P04. There were no *A. fatua* types in the *A. byzantina* cluster. Although these species do not have substantial representation, collection sites and positions on the MDS space (Supplementary Fig. 7) suggest that *A. occidentalis* and *A. hybrida* have regional origins in West Africa and Iran (respectively) while *A. fatua* is cosmopolitan without an obvious center of origin. The fatuoid (floret shattering) character that defines these three species has long been understood to be controlled by a single recessive genetic factor[6], however, it has not been resolved whether this character arises through mutation, chromosome

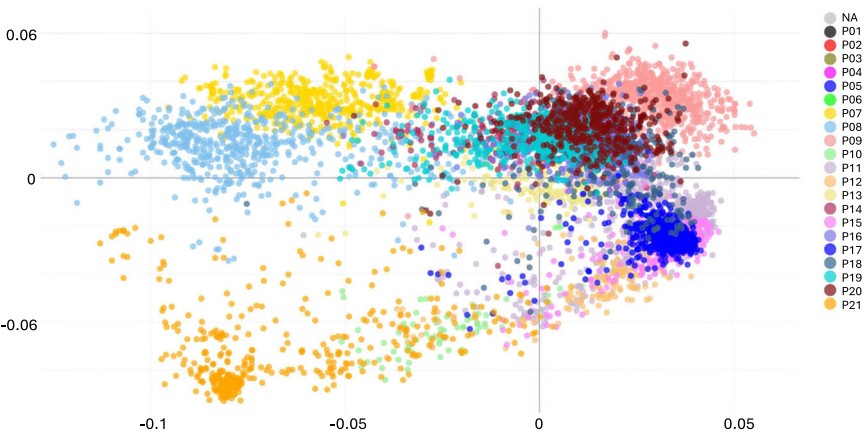

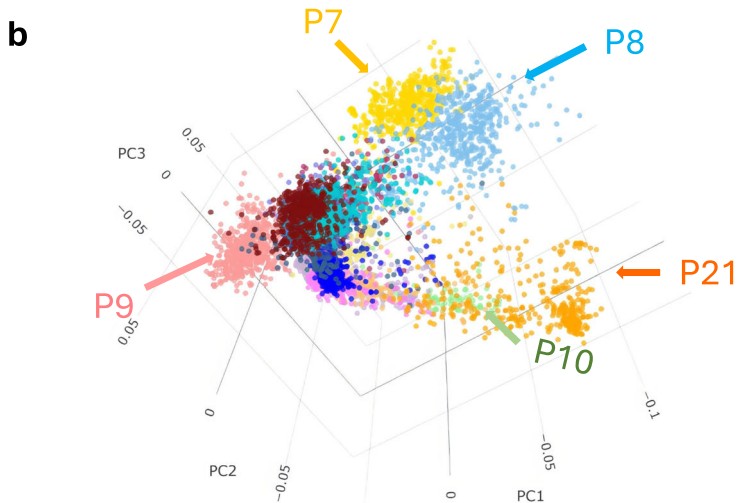

**Fig. 2 | Populations of cultivated oat.** Multi-dimensional scaling of $n = 6928$ taxa from populations containing primarily cultivated *A. sativa* (**a**) shown on axes 1 and 2 and (**b**) shown on a projection of the first three axes. Membership in applicable populations from $K = 21$ is colored as in Fig. 1. Populations P07 (Australian oat), P08 (southern USA), P21 (winter oat), and P09 (North Dakota type) are indicated, as discussed in the text. Online interactive versions of these plots can be used to visualize or highlight additional populations or features (https://graingenes.shinyapps.io/Oat_diversity/). Source data are provided as a Source Data file.

aberration, or hybridization. Our observation of *A. fatua* accessions within multiple populations of cultivated oat suggests that many of these lines have a recent and spontaneous origin. Once established, these fatuoid shattering lines could find local or temporary selective advantage within cultivated crops.

**Patterns of cultivated oat genomic diversity**

Except for the Spanish landraces (P06) the populations containing cultivated *A. sativa* are less distinct than those of *A. sterilis* and *A. byzantina*. To explore cultivated oats in more detail, we performed an MDS analysis that was restricted to populations P07 through P21 (Fig. 2). This allowed better visualization of the cultivated *A. sativa* populations, especially by rotating or selecting additional axes on the live version of the MDS plot (https://graingenes.shinyapps.io/oat_diversity/). We then compared Fig. 2 with heatmaps showing population membership by country (Fig. 3a) or North American state or province (Fig. 3b).

Populations P07, P08, P10, and P21 were separated effectively from the remaining populations on the first axis, while the second axis separates P10 and P21 from most other populations. Populations P07 and P08 are representative of Australia and southern USA, respectively, with P08 also containing some lines from Brazil and Argentina. In these regions, oat is produced during the winter season (i.e., fall sown, winter grown), where vernalization is not required, but daylength insensitivity is essential. Population P10 contains landraces from Turkey, and P21 contains lines that are representative of regions where oat is grown as a true winter crop (sown in the fall and dormant through the winter), including the UK and Virginia, USA. The third axis (Fig. 2b) more effectively separates population P09 from the others. This population originates primarily from breeding programs in Canada and northern USA states, in particular North Dakota (Fig. 3b).

Oat varieties originating from North America were distributed more evenly across different populations than those from other countries, which were often dominated by only one or two populations

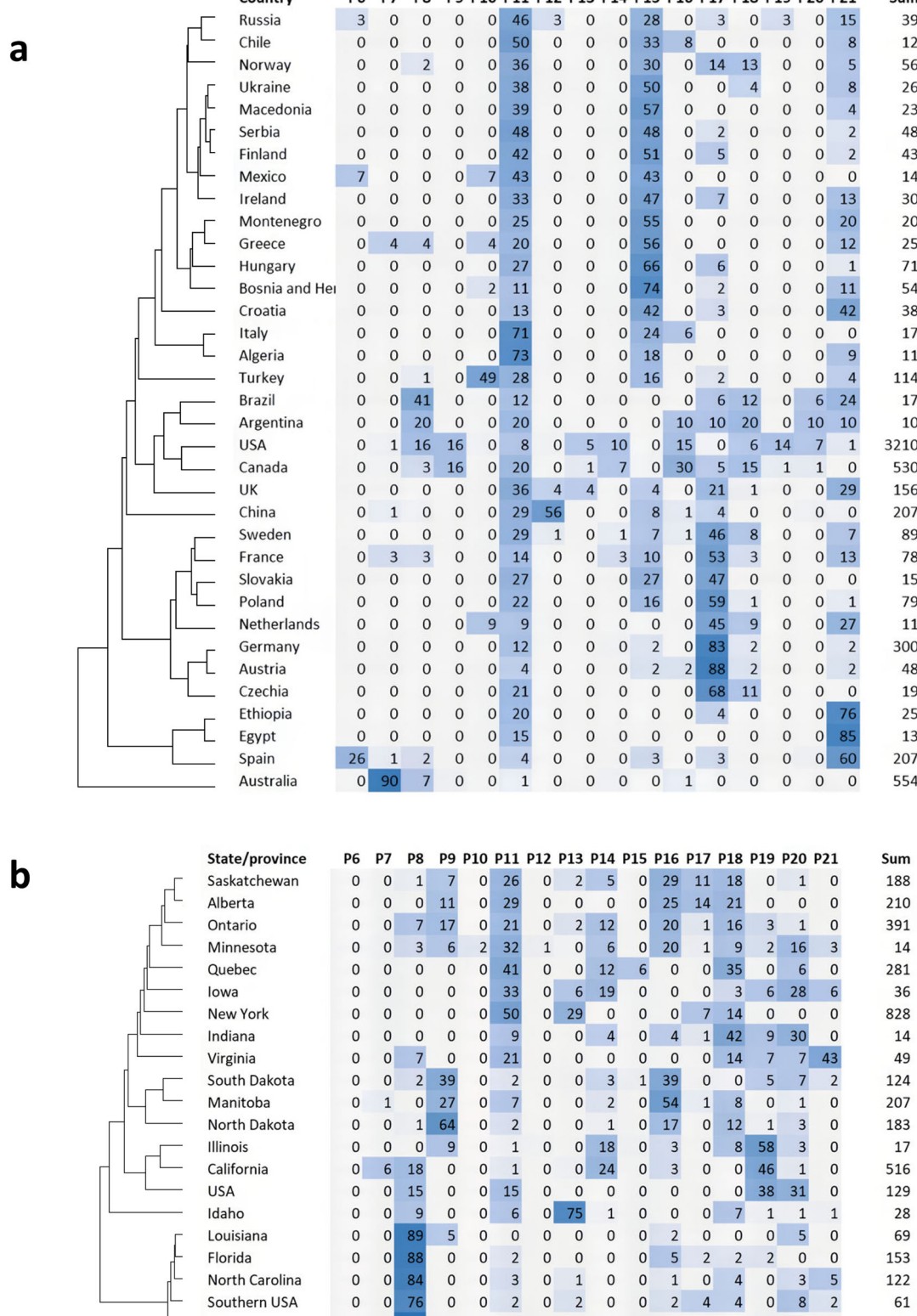

**Fig. 3 | Heatmaps of $K$ = 21 sparse nonnegative matrix factorization (sNMF) populations of *A. sativa* taxa. a** Population membership in each of $K$ = 21 sNMF populations summarized by country. **b** Population membership summarized by North American state and province. Numbers indicate percentage membership of a country, state, or province in a given population. Light blue indicates a low percentage (2–10%) and dark blue indicates a high percentage (over 50%) while no color indicates 0–1%. Populations without taxa are not shown. Countries or states with fewer than ten accessions are omitted. The sum is the total number of taxa in each row. Percentages do not always sum to 100 due to rounding. Source data are provided as a Source Data. file.

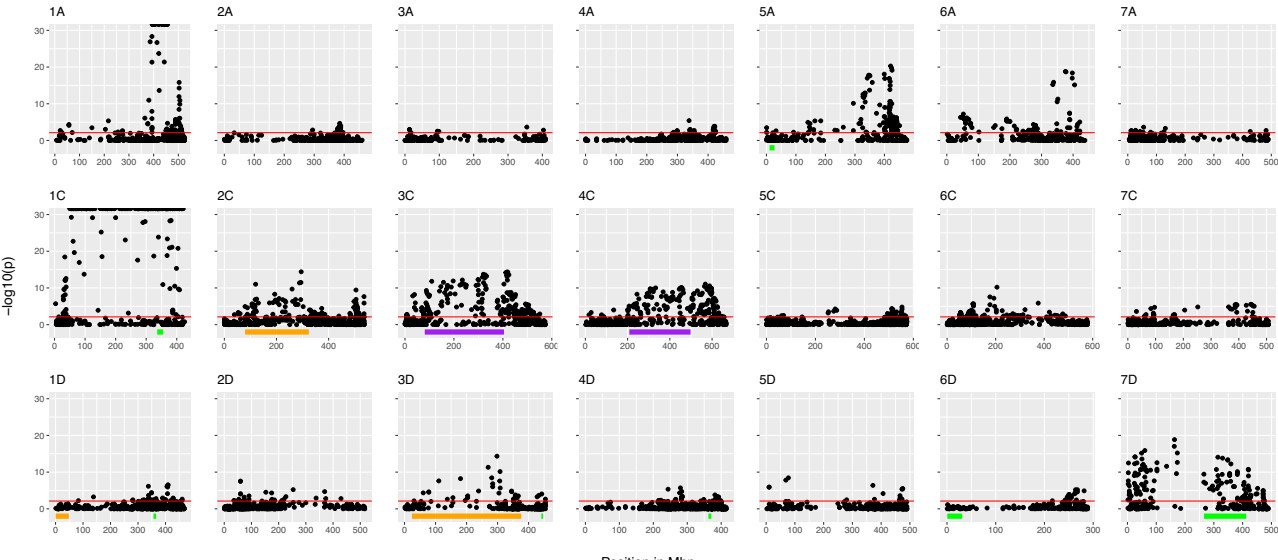

**Fig. 4 | Impact of chromosome rearrangements on hexaploid *Avena* population structure and local adaptation revealed by complementary population genomic approaches.** Manhattan plots show the PCA-based genome-wide scans of local adaptation across the 21 chromosomes, with chromosome names shown on top of each panel, analyzed using PCAdapt for the full set of taxa. The *y*-axis represents the $-\log10(p)$ values of marker associations with population structure. This *y*-axis is truncated, such that highly significant points are crowded at the top. The red line at $y = -\log10(p) = 2.12$ marks the false discovery rate threshold lower than 5% (adjusted for multiple comparisons) for detecting outlier loci (Supplementary Fig. 9). Shaded areas below zero and above the *x*-axis labels indicate putative inversions identified by Lostruct analysis. The colored bars correspond to the outlier genomic regions (green for corner 1, orange for corner 2, and purple for corner 3) in Supplementary Fig. 10. Source data are provided as a Source Data file.

(Fig. 3a). While some of this may be due to sampling bias, there appears to be a greater genetic diversity of germplasm in North America than in Central and Northern Europe and China (Fig. 3a and Supplementary Fig. 8).

## Large-scale chromosome variations related to population structure and adaptation in *Avena*

Large-scale chromosome translocations and inversions are common in oat, and these may shape population structure, adaptation, speciation, and domestication[36]. This study applied two complementary population genomics approaches to investigate structural variations in *Avena* species. The first was a PCA-based genome-wide scan for local adaptation (PCAdapt) used previously to identify outlier loci[37] that were subsequently confirmed to be related to translocations and inversions[8,38]. To complement this, we employed the Lostruct method, which detects putative inversions by identifying abnormal patterns of genetic relationship in non-overlapping chromosome windows[36].

The genome-wide PCAdapt scan identified 1769 outlier loci distributed across all chromosomes (Fig. 4 and Supplementary Fig. 9). Seven chromosomes (1A, 1C, 2C, 3C, 4C, 5A, and 7D) each had more than 120 outlier loci, many of which were associated with known translocations and inversions. Many of these regions correspond to confirmed structural differences among pangenomes, including inversions on chromosomes 3C, 4C, and 7D, and translocations involving chromosome pairs 1A/1C, 1D/6C, and 2A/2C. The combined effect of the 1A/1C translocation and the small putative inversion on 1C may explain the high number ($n = 301$) and distribution of outlier loci along this chromosome and the high concentration of outlier loci near the end of chromosome 1A ($n = 125$).

The Lostruct analysis identified 27 outlier windows at three corners of the MDS plot (Supplementary Fig. 10). As expected, these regions tended to overlap with chromosome inversions (including ones on 3C, 4C, and 7D) but not with chromosome translocations. Of the 27 inversion windows detected by Lostruct, only two, on chromosomes 6D and 1D, did not overlap with PCAdapt outlier regions

(Fig. 4). These results may be affected by marker density and/or polymorphism content. For example, low marker density on chromosome 7D may be the reason that the detected outlier window (256.6–412.1Mbp) is smaller than the size of the physical inversion.

It is beyond the scope of this study to characterize all of the putative inversions and translocations. Instead, we focus on two key structural variations: a recently confirmed inversion on chromosome 7D and the well-known 1A/1C translocation. Both of these play significant roles in oat population structure, ecotype differentiation, and local adaptation, as documented in the companion work and earlier studies[8,37–39].

## Chromosome 7D

*K*-means clustering of PCA results of 150 SNPs from the outlier window on chromosome 7D revealed four haplotypes (7D-H1 through 7D-H4; Supplementary Figs. 10 and 11a, b). An LD analysis on chromosome 7D (Supplementary Fig. 11c, d) showed less LD within the 7D-H3 carrier group compared to the full set. Moreover, recombination on chromosome 7D was suppressed in crosses between parents with contrasting haplotypes (specifically 7D-H3 *vs.* 7D-H1) compared to crosses between parents that carried the same haplotype (Supplementary Fig. 11e, f). These results corresponded to confirmed chromosome arrangements in a number of reference genomes. *A. byzantina* PI258586, *A. sterilis* TN4, "Gehl", "Bannister", "Bilby", and "Williams" contain 7D-H1, while "Victoria" contains 7D-H2. These accessions all contain a non-inverted ancestral chromosome configuration on chromosome 7D. In contrast, *A. sterilis* TN1, "HiFi", "Leggett", "AC Morgan", and other genomes having the inverted configuration of chromosome 7D[32] all contained haplotype 7D-H3. The presence of these haplotypes was verified using whole genome resequencing data (WGS)[32], with the GBS and WGS datasets showing 96.2% concordance. Avni et al.[32] also showed that lines with a non-inverted 7D chromosome haplotype were associated with earlier heading compared to those with the inverted haplotype, suggesting that this inversion and the associated haplotypes play a role in adaptation.

The distribution of the four 7D haplotypes across hexaploid oat species and within each of the $K = 21$ populations is shown in Fig. 5. The presence of different 7D inversion states in *A. sterilis* is suggested by the contrasting haplotype membership across the four *A. sterilis* populations (P02 to P05). We hypothesize that P05 represents a primary founding population of *A. sativa*, and this is supported by the predominance of the inverted haplotype 7D-H3 in both P05 and *A. sativa*. The mixture of the non-inverted haplotypes 7D-H1 and 7D-H2 in both P02 and *A. byzantina* supports our hypothesis that P02 is a founding *A. sterilis* wild population of *A. byzantina* (P01).

Within *A. sativa*, the prevalence of the two non-inverted (7D-H1, 7D-H2) haplotypes in winter/autumn-sown oat populations (P07 and P08) is consistent with the historical use of *A. byzantina* crosses in the associated breeding programs. Population P20, prevalent in South American countries with a similar production system, also appears to contain a mix of inverted and non-inverted haplotypes. In contrast, most European, UK, and North American winter types in P21 carry the inverted 7D-H3 haplotype. Founder lines of many varieties in P21 are the historical landraces "Winter Turf" or "Virginia Gray"[40]. The Winter Turf sample in this study belongs to P21 and carries 7D-H3. These patterns may be related to flowering time and daylength sensitivity loci found on chromosome 7D[38], and it further highlights the need to distinguish between fall-sown oat varieties that mature during the winter *vs*. those that are dormant in the winter. The relatively small proportions of 7D-H1 and 7D-H2 among most other *A. sativa* populations probably resulted from the exchange of oat-breeding germplasm among breeders in different regions.

Interestingly, the rarest chromosome 7D haplotype (7D-H4) is present primarily in some of the Turkish landraces within P10, as well as most of the Chinese hulless landraces in P12. Despite its rarity in cultivated oat populations, 7D-H4 is also found in a small number of accessions of *A. fatua*, *A. hybrida*, and *A. sterilis*.

### Chromosome 1A/1C

To investigate the chromosome 1A/1C translocation, we applied *K*-means clustering based on PCA of 532 SNPs in the translocated region (chromosome 1A, 420.2–525.3 Mbp) of the Sang reference genome, and inferred five distinct haplotypes (Supplementary Fig. 12). The distribution of these haplotypes within oat germplasm revealed patterns, many of which are consistent with the chromosome 7D haplotypes described above. Notably, however, the 1AC-H1 haplotype was unique within P09, and not found in any potential founding populations. This may account for the genetic uniqueness of P09 in the MDS analysis (Fig. 2) and suggests a possible origin through introgression from an exotic source. One suspected origin is the synthetic line 'Amagalon' (*A. magna* x *A. longiglumis*) which has been used in the North Dakota oat breeding program. The existence of a secondary or compound 1A/1C translocation in Moroccan *A. occidentalis* genotype CN 25955[32], as well as a 1C/1D interchange described three decades ago in the "Kanota" monosomic series[41], suggests occasional homoeologous pairing might be operative in creating novel haplotypes within the *Avena* Group 1 chromosomes.

### Diversity of the oat pangenome

In Supplementary Discussion 3 and Supplementary Method 3, the 29 hexaploid reference genomes sampled in the oat pangenome project were projected onto the MDS diversity space (Supplementary Fig. 13) and the 21 sNMF populations (Supplementary Table 3). All populations contained at least one reference genome except for *A. sterilis* population P02, and *A. sativa* populations P08, P10, P18, and P19. These results and future implications are discussed further by Avni et al.[32].

## Discussion

This large-scale diversity analysis across all major hexaploid *Avena* species combines data from 15 GBS experiments with varieties and accessions from diverse locations contributed and interpreted by collaborators from around the globe.

Our results show a high degree of structure in four populations of *A. sterilis* that is partially associated with collection site (Fig. 1). Given the increased sampling depth and marker density, we can now resolve more definitively questions that were previously unanswered and propose avenues for directed research. For example, Goffreda et al.[25] found significant divergences between accessions from Iran and Iraq relative to accessions from more western regions around the Mediterranean Sea. We can now classify two divergent populations of *A. sterilis* (P02 and P05) present in Iran and Iraq, both of which contrast with populations P03 and P04 from the eastern and western Mediterranean, respectively.

Surprisingly, there is also a large overlap among different *A. sterilis* populations within relatively small regions, such as eastern Turkey and Jordan. Further investigations may elucidate whether local conditions have influenced this population divergence. However, several structural chromosome variants in addition to the 1A/1C translocation have been suggested by this work and/or confirmed by WGS. In particular, the chromosome 7D inversion differentiates two *A. sterilis* lines (TN1, inverted *vs*. TN4, non-inverted) that represent populations P05 *vs*. P03, respectively. While further sequencing of accessions from different *A. sterilis* populations is required, it now seems likely that structure-associated reproductive barriers or genetic islands may have helped to preserve the genetic distinctness of *A. sterilis* populations having different adaptations. Such phenomena are documented in other plant species, including sunflowers[36] and barley[42].

Our findings on the divergence of *A. sterilis* (Fig. 1) also open more evolutionary questions. Did these divergences arise through mutation of a common lineage, or did they arise from separate polyploidization events? For example, the wheat D genome was contributed by different lineages of *Aegilops tauschii*[43]; thus, it is possible that different populations of *A. sterilis* might also originate from divergence among ancestral oat diploids or tetraploids prior to polyploidization. It is also likely that a more recent redistribution of wild oat populations was assisted by deliberate cultivation as a primitive forage species, as is still observed[44], or that early domestication of oat as a grain species occurred outside of the original geographic range of the founding population, as has been reported in rye[45].

Our results support the conclusions of Canales et al.[22] and Nikoloudakis et al.[23], who determined that oat landraces and varieties classified as *A. byzantina* are highly distinct from *A. sativa*, and justify their classification in a separate taxon. *A. sativa* is usually distinguished from *A. byzantina* by a major 1A/1C chromosome translocation present in most *A. sativa* varieties. This translocation introduces pseudo-linkage and reproductive barriers when carriers are crossed with non-translocated types from *A. byzantina*[38]. Crosses between *A. byzantina* and *A. sativa* have been made for mapping studies[46] but are less commonly made for cultivar development. A notable exception is the use of red-seeded oat varieties such as 'Fulghum', Victoria, 'Red Algerian', and Kanota in cultivar development. Accessions and varieties derived from these red-seeded oats can be found at different frequencies within the various *A. sativa* populations identified by our analyses, likely accounting for the presence of the non-translocated 1A/1C chromosome configuration in some cultivated *A. sativa* varieties.

We can now reason that the translocated *A. sativa* type was domesticated from an *A. sterilis* population that contained the translocation, while *A. byzantina* was domesticated from a different *A. sterilis* population without the translocation. We speculate, based on genetic proximity, that the founding *A. sterilis* groups of cultivated *A. sativa* and *A. byzantina* may have originated from P05 and P02, respectively. These origins are further supported by our analysis of local haplotypes related to the inversion on chromosome 7D (Fig. 5)

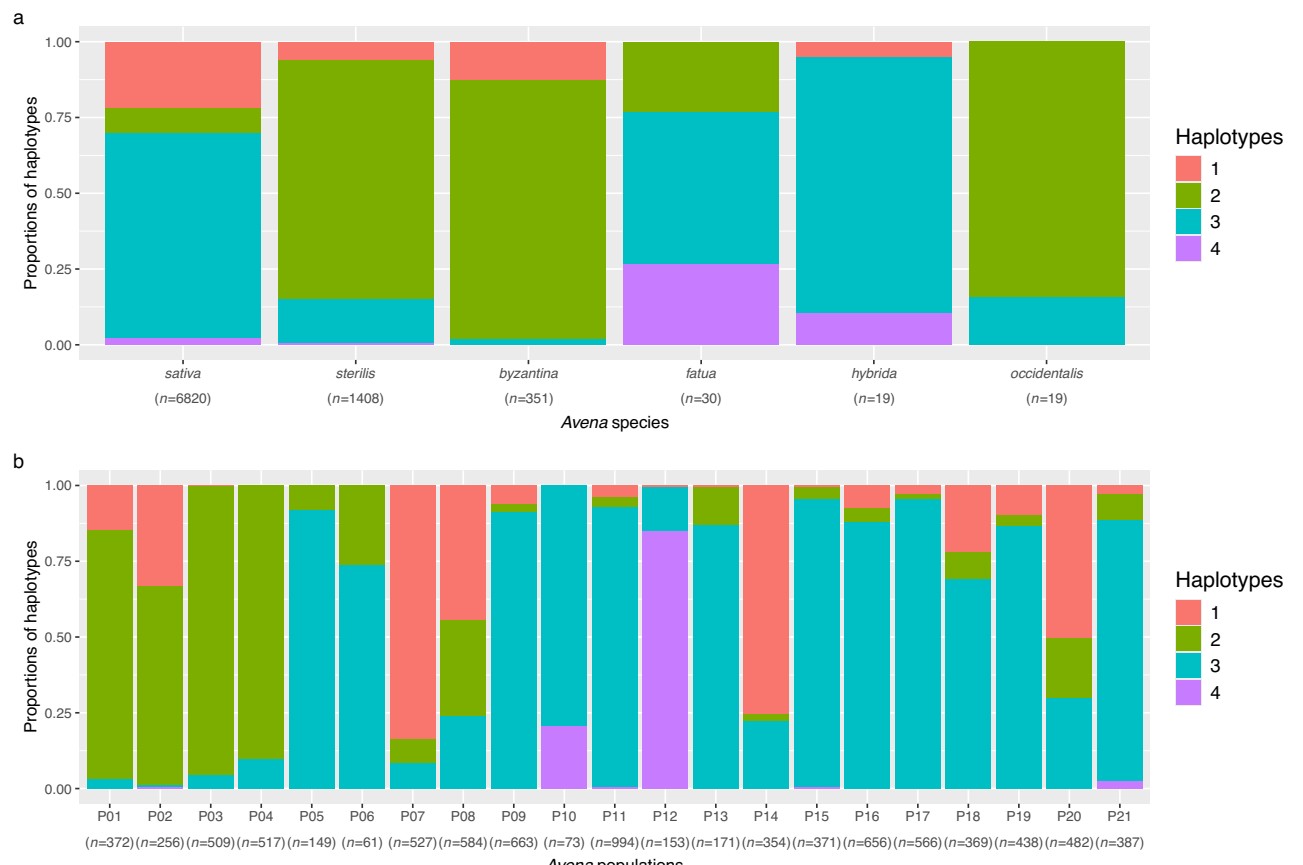

**Fig. 5 | Chromosome 7D inversion karyotypes in hexaploid *Avena* reveal population structure and patterns of agroecological adaptation. a** The proportions of four putative haplotypes (as revealed by *K*-Means clustering) in each of six hexaploid species. The four haplotypes are haplotype 1 (7D-H1) ($n = 1638$), haplotype 2 (7D-H2) ($n = 1983$), haplotype 3 (7D-H3) ($n = 4868$), and haplotype 4 (7D-H4) ($n = 133$). **b** The proportions of the four haplotypes in each of the 21 *Avena* populations (P01 to P21). Source data are provided as a Source Data. file.

and the translocation on chromosomes 1A/1C (Supplementary Fig. 12). Whole-genome sequencing experiments have confirmed the presence of the translocation in *A. sterilis* TN1, belonging to P05, but an accession from P02 has not yet been sequenced.

Is *A. sativa* a mosaic of multiple origins, or did it arise primarily from P05 of *A. sterilis* with occasional introgressions? Based on genetic proximity (Fig. 1a), we speculate that P05 was the primary founding population of certain Spanish landraces (P06), Turkish landraces (P10), and cultivated white winter oat (P21). It follows that other *A. sativa* populations arose from these early domesticated and genetically proximal landraces, rather than through independent domestication. However, modern oat cultivars, especially those from North America, appear to have diverged extensively compared with those from Europe. It may be that North American oat cultivars are more influenced by crosses with *A. byzantina*, *A. sterilis*, or non-hexaploid species that have been used to introduce traits such as cold tolerance and pathogen resistance. As an example, population P09 is heavily influenced by oat cultivars originating from the breeding program at Fargo ND, USA. It also includes the cultivar HiFi, which has the synthetic hexaploid line Amagalon in its pedigree. The frequent use of HiFi as a parent to introduce rust resistance and increase fiber content of Canadian and Northern US oat varieties may account for the expansion of population P09. Other introgressions of rust resistance, summarized recently by Park et al.[47], may have driven other differences in population structure.

Apart from biological drivers of diversity, cultural and historical differences may also be relevant. Such factors may then be preserved through other mechanisms. For example, differences between northern and southern cultivars from the USA may reflect the distinctive human migration patterns during European colonization, with the southern USA lines being genetically close to Spanish lines already adapted to the Mediterranean climate and production[13,22]. While there are probably only a small number of genes with major effects on flowering time and pathogen resistance, breeders will generally avoid crossing with material that falls far outside their region of adaptation, except when it is necessary to introgress a new trait such as disease resistance. Moreover, it was shown recently that the chromosome 7D inversion causes segregation distortion in a typical north-by-south population[38], and such chromosome rearrangements may also be responsible for maintaining clusters of adaptation related loci, even when wide crosses are made. Other chromosomal rearrangements (Fig. 4), which now appear to be more widespread than previously suspected, may play similar roles.

The relatively high diversity of North American cultivars compared to the sampled European cultivars may have resulted from several factors. There are more oat breeding programs in North America than in Europe, and North America has a wider range of climates and daylengths, different regional pathogen pressures, and different regional preferences for grain quality parameters. The genetic and phenotypic diversity of oat populations in the Americas

highlights the significant role that plant introductions and wide crosses have played in facilitating this diversity[40].

The complementary use of pangenomics and population genomics enabled us to perform in silico karyotyping (e.g., Figs. 4, 5, and Supplementary Fig. 12) and to infer the otherwise hidden impact of chromosome rearrangements on adaptation and ecotype differentiation. It also helped to explain the presence of large haplotype blocks and local segregation distortion in a genomic region containing quantitative trait loci for disease resistance and flowering time. A similar approach, reported in rice, identified lineage-specific haplotypes for traits and helped explain the genomic organization of rice species[48]. While the chromosome 1A/1C translocation and the chromosome 7D inversion are now well documented and characterized, inversions on chromosomes 3C and 4C are confirmed but not yet characterized[32]. Other large-scale inversions on chromosomes 3D and 2C are postulated only by our population genomics approach but have yet to be confirmed by genome sequencing. In our companion paper, we show an example of the co-occurrence of the 1A/1C translocation with postulated 2C, 3C, and 7D inversions, resulting in distorted segregation and suppressed recombination on multiple chromosomes. It is clear from this work that chromosome rearrangements are integral to the origin, domestication, differentiation, and cultivation of oat, and that further studies and diagnostic tools to identify these phenomena may have important applications in oat improvement.

We have already discussed the hypotheses for a separate domestication of cultivated *A. sativa* vs. *A. byzantina* based on genomic proximity. The large size of the chromosome 7D inversion, and the presence there of adaptive loci related to daylength sensitivity, identify chromosome 7D as both an indicator and a driver of domestication events. We now hypothesize three potential domestication pathways characterized by chromosome 7D haplotypes. (1) Cultivated *A. byzantina*, with a mixture of non-inverted 7D-H1 and 7D-H2 haplotypes, was domesticated from *A. sterilis* P02, having a similar haplotype mix. One or both of these haplotypes may confer daylength insensitivity, resulting in the use of *A. byzantina* for winter (short day) production in Mediterranean climates. (2) Cultivated *A. sativa*, the majority of which contains the inverted 7D-H3 haplotype, was domesticated from *A. sterilis* P05, which is the only *A. sterilis* group containing a majority of this haplotype. We do not know yet if daylength sensitivity originated within *A. sterilis*, but this would have been an important domestication event within this haplotype that facilitated the expansion and adaptation of oat to northern regions. While we consider populations P07, P08, P14, and P20 to be *A. sativa*, they also carry large proportions of the 7D-H1 and 7D-H2 haplotypes, resulting from crosses with *A. byzantina* in their breeding history (possibly preserving daylength insensitivity), as well as other *A. byzantina* introgressions, such as the non-translocated 1A/1C haplotypes described in hypothesis 1. (3) A third hypothesis, related to an independent domestication of Turkish and Chinese hulless landraces (P12 and P10) carrying 7D-H4, is also supported by a recent domestication study[21]. This haplotype is rare in the *A. sterilis* populations that were sampled, but it may have arisen from weedy species, such as *A. fatua*. We note, however, that the gene conferring hullessness is not located on 7D, and that our diversity study does not support the inference of extensive diversity and distinctness of Chinese landraces made by Nan et al.[21].

These hypotheses have important biological and evolutionary implications that warrant further testing. However, domestication is not a single event confined to a single location but, rather, a complex and gradual phenomenon that occurs over a long period, often involving multiple populations and geographic regions[49]. Thus, many questions remain regarding the domestication and demographic history of the cultivated hexaploid oat. Further germplasm collection and/or sampling of existing collections of *A. sterilis* are needed to address these evolutionary questions and to inform future breeding efforts.

While all hexaploid *Avena* species are considered part of the primary gene pool[2], breeders have only attempted to exploit the vast genomic and phenotypic diversity present in wild *Avena* species when absolutely necessary (e.g., to incorporate resistance to devastating diseases). Since crosses among hexaploid oat species are usually fertile, this reluctance is primarily due to the presence of deleterious alleles linked to desirable traits from wild relatives, a phenomenon known as linkage drag. Breeders can reduce linkage drag through marker-assisted introgression. However, this process is time-consuming and frequently hampered by segregation distortion and recombination suppression. A marker-based structural variant detection method (in silico karyotyping, as described here) could facilitate the selection of optimal parental line combinations[50], reducing the impact of linkage drag by promoting recombination during the introgression of disease resistance or other desirable traits from species like *A. sterilis*[47]. This method also holds potential for introducing quantitative traits, such as adult plant resistance to crown rust or other agronomic and adaptation-related traits. For example, in potatoes, it was demonstrated that inversion maps and pangenome information could enhance parental selection for efficient introgression of carotenoid content in tubers by precluding a 5.8 Mb inversion containing 464 genes closely linked to a gene of interest[51].

In conclusion, we analyzed and presented these data to examine the worldwide diversity and population structure of hexaploid oat, and to support the development and testing of further hypotheses. These results and the associated data sets will encourage and enable further exploration by specialists in different domains. For example, it is now possible for breeders to identify populations that represent their breeding material and, potentially, to expand diversity through crosses with taxa from diverse populations having similar adaptations and in silico karyotypes. Future experiments, such as the development of an extended oat pangenome, can now prioritize the selection of taxa from under-represented populations carrying key structural variants, as identified in this study. These future studies should include the analysis of populations relative to structural chromosome variants, broad-based analyses of linkage disequilibrium, trait association studies, and detailed analysis of functional genes to ascertain causes and consequences of domestication and oat genome evolution. These findings and genomic tools could inform targeted breeding strategies to develop climate-resilient oat varieties with improved yield stability under changing environmental conditions.

## Methods

### SNP calling and data filtering

Published and unpublished GBS data from oat diversity experiments were assembled and curated (Table 1). Each unique operational taxonomic unit (hereafter, a taxon, accession, variety, or breeding line, depending on context) was assigned a unique ID (GBSID) and all measurements were taken from distinct samples. The full list of taxa and available metadata are provided in Supplementary Data 1. All data were produced using GBS methods similar to those described by Huang et al.[30]. Briefly, DNA samples were prepared from single plants derived from single seeds randomly selected from each accession. The GBS libraries were constructed in 96-well plates where they were double-digested using restriction enzymes PstI (CTGCAG) and MspI (CCGG) and ligated to barcoded adapters that were unique to each sample within a plate. The DNA barcodes contained the overhang generated by PstI restriction digest, and ranged in length from 4 bp to 9 bp, balancing the base composition of the GBS library. For each plate, a single random blank well was included for quality control to ensure that libraries were not switched during construction, sequencing, and analysis. Samples then were pooled by plate into libraries and

polymerase chain reaction-amplified. Barcodes, plate identifiers, and plate coordinates are provided as Supplementary Data 5. Barcoded GBS libraries were sequenced by the methods shown in Supplementary Table 1.

Data were analyzed using a reference-based GBS approach. Reads were de-multiplexed into separate fastq-formatted files for each accession. Barcodes were removed so that each read began with the 5′ *PstI* site, while the *MspI* site and adapters were left for trimming later in the pipeline. De-multiplexed data were submitted to the European Nucleotide Archive (Project ID PRJEB63645 with individual sample IDs provided in Supplementary Data 1. Submissions were made for all datasets utilized in this study, even if previously published elsewhere, to ensure data uniformity and provide a single centralized source for future research.

A primary set of SNP calls was generated using methods similar to Milner et al.[52]. Reads were aligned to the Sang V1 reference genome[8] available from ENA under accession PRJEB44810. Variant analysis was performed by aligning reads to the reference using BWA-mem[53]. The BAM files were then sorted using Picard[54], followed by SNP calling using a SAMtools/bcftools[55] script that split the chromosomes into 5 Mb fragments or smaller. The results were concatenated using the bcftools "concat" function. The initial SNP filtering was conducted with a custom script (https://bitbucket.org/ipk_dg_public/vcf_filtering) using the following parameters: minimum depth for homozygote = 2, minimum depth for heterozygote = 4, minimum SNP quality = 40, minimum sequencing depth at a SNP site = 100, minimum minor allele frequency = 0.01, and minimum presence = 0.5. Biallelic SNPs with a maximum of 50% missing values were then filtered using vcftools[56].

Secondary filtering at two levels of stringency was performed using TASSEL 5.2.94[57]. A large matrix (Matrix50) was filtered for taxa with ≥2% non-missing genotypes, followed by genotypes with ≥50% non-missing taxa. A smaller and more stringent matrix (Matrix80) was filtered first for taxa with ≥10% non-missing genotypes, then for genotypes with ≥80% non-missing taxa. Matrix50 provides the community with a larger set of markers and taxa, while Matrix80 was used for analyses that were more sensitive to missing genotype data, as described below. Genotypes in both matrices were also filtered to remove sites with minor allele frequency ≤1% and heterozygosity ≥5%. Missing genotypes in Matrix80 were then imputed by the method of linkage disequilibrium (LD) KNNi[58] using default TASSEL parameters (high LD sites = 30, number of nearest neighbors = 10, max distance between sites to find LD = 1E7). Filtered genotype data were saved as compressed HapMap files for further analysis.

## Multi-dimensional scaling and population structure analysis

Matrices of distance among taxa were generated in TASSEL using the default method, which computes distance (D) between each pair of taxa as the proportion of non-identical sites among all non-missing genotypes. Distances were written to compressed square matrices for further analysis. Classical (metric) multi-dimensional scaling (MDS) of taxa was computed from the distance matrices using the cmdscale routine in the R statistical software package (R version 4.3.2)[59]. For comparison, MDS and PCA were both conducted using default routines in TASSEL. The imputed marker dataset, with 8,816 taxa (excluding the low-read Jordanian samples) and 19,928 sites, was used for the downstream population structure-related analysis. Population structure analysis was performed using the imputed data with sparse nonnegative matrix factorization and a least-squares optimization algorithm (sNMF)[60] as implemented in R, and K-means clustering using the ADEgenet[61] packages in R.

## Chromosome regions associated with population structure or adaptation

To identify genomic regions associated with population structure and local adaptation, we applied the Mahalanobis distance method within the PCAdapt[62] package in R to analyze the first 14 principal components. Significant markers and genomic regions were identified after correction for a false discovery rate (gamma = 0.05). To identify genomic regions associated with inversions or recombination suppression, local structure analysis (Lostruct)[63] was run using the default 50 SNP window settings. To reduce the computational demand for Lostruct analysis, accessions with more than 50% missing genotypes after imputation were removed, resulting in 8652 taxa. To characterize outlier regions identified through Lostruct, we performed PCA and K-means clustering on markers within these regions. SNPs were filtered using SNPrelate[64]. For example, 150 SNP markers from 256.6Mbp to 412.1Mbp were used for the PCA analysis of the 7D outlier region. To identify distinct clusters within outlier regions, we used the KMeans method[65] in R. Clusters were determined based on the first principal component's minimum, maximum, and median values. The proportion of between-cluster sum of squares over the total was used as the measure of discreteness for classification. For example, in the analysis of the chromosome 7D inversion, multiple levels of clusters (K) were tested, and the minimum number of clusters (K = 4) that gave a ratio above 0.9 (K = 4; 0.94) was selected for further investigation. Haplotypes were abbreviated by chromosome followed by numeric identifier (e.g., 7D-H1). To assess LD patterns within Lostruct outlier genomic regions, we conducted LD analysis using PLINK[66] with default parameters, using all genotypes carrying the four haplotypes (n = 8652) compared to those from the largest haplotype group (7D-H3; n = 4863). To validate the identified chromosomal rearrangements, we analyzed progenies of biparental mapping populations available to the research team. For example, the "Goslin" x "HiFi" (n = 160) and "TX07 CS-1948" x "Hidalgo" (n = 515) populations[38] were used to validate our detection method for the 7D inversions on recombination fraction[37,38].

## Online visualization

A dynamic, web-based application was developed using R-shiny[67] to display and rotate two- or three-dimensional MDS plots of the taxa, overlain with customizable colors representing taxa metadata or population membership. Collection sites for taxa (Supplementary Data 1) from non-cultivated species and some *A. byzantina* samples were positioned on an interactive map using Google Maps.

## Reporting summary

Further information on research design is available in the Nature Portfolio Reporting Summary linked to this article.

# Data availability

The de-multiplexed sequencing data are available in the European Nucleotide Archive under accession PRJEB63645 with individual sample IDs provided in Supplementary Data 1. All metadata for accessions used in this study are available as Supplementary Data 1. All SNP data from Matrix50 (unfiltered) have been deposited to both the T3/Oat Database [https://oat.triticeaetoolbox.org/breeders/trial/6768] with an associated browsable online dataset [https://divbrowse.triticeaetoolbox.org/index.html], as well as the European Variation Archive (Project: PRJEB86345 [https://www.ebi.ac.uk/ena/browser/view/PRJEB63645]; Analyses: ERZ25059875). The filtered/imputed SNP data matrix (Matrix 80) is available as a downloadable file from GrainGenes (see link below). Data files and links to online resources are also available on our project site on the GrainGenes[33] database [https://graingenes.org/GG3/content/global-oat-genomic-diversity-project]. Source data are provided with this paper.

# Code availability

Custom R scripts that were used to analyse this project are provided in Zenodo [https://zenodo.org/records/14976680]. The code and sample files for the program ParseBlastSNP (used in Supplementary

Discussion 3) are available on the Zenodo repository [https://zenodo.org/records/14889743] where they are provided as open source software with no restrictions on use. Both items are located on Zenodo project page [https://zenodo.org/communities/god/].

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

## Acknowledgements

W.B., A.I., C.W., J.B., J.M.F., K.N., I.P., W.Y., A.D., Y.F., and N.T. acknowledge that their work was supported by Agriculture and AgriFood Canada (AAFC) and by partnership funding contributed by the Canadian Field Crops Research Alliance and the Prairie Oat Breeding Consortium. WB and NT acknowledge co-funding by AAFC and Genome Quebec in the "TUGBOAT" project. C.B., S.M., R.N., J.F., R.G., K.K., J.J., and T.Se acknowledge that their work was supported by the U.S. Department of Agriculture, Agricultural Research Service. J.I.S. was supported by the Severo Ochoa Program for Centres of Excellence in R&D from the Agencia Estatal de Investigación of Spain, grant SEV-2016-0672 (2017-2021) to the CBGP, and by grant PID2021-123718OB-I00, funded by MCIN/AEI/10.13039/501100011033 and by "ERDF A way of making Europe," CEX2020-000999-S. E.Pr. and F.C. were supported by grant [PID2022-142574OB-I00] funded by MICIU/AEI/10.13039/501100011033, FEDER, UE, and Junta de Andalucia [QUAL21_023 IAS]. C.H. and T.L. acknowledge support by BBSRC grants BBS/E/IB/230001, BBS/E/W/0012843, BB/M028151/1 and BB/S008195/1. ID acknowledges support from the California Crop Improvement Association (CCIA) and the NACA-Oat from USDA. Y.P. acknowledges support from the Science and Technology Plan of Sichuan-China (No. 2024NSFJQ0003). R.A. acknowledges that his work was supported by the Alexander von Humboldt foundation. S.B. and M.H. acknowledge that their work was supported by the German Federal Ministry of Food and Agriculture within the framework of the Federal Organic Farming Programme (grant number 28AIN02A20). P.Z. acknowledges support from the Grains Research and Development Corporation, GRDC, Australia.

## Author contributions

Planned this study: W.B., M.Ma., N.T.; collected and contributed data: A.Be., A.D., A.I., C.C., C.H., C.W., E.Pr., F.C., I.D., I.P., J.A., J.M.F., R.G., R.N., J.H., J.I., J.J., J.M., K.K., K.N., K.S., L.G., M.C., M.H., M.Mc., N.T., P.Z., R.N., S.B., S.H., T.L., T.Su., W.Y., Y.F., Y.H., Y.L., Y.P.; coordinated and designed public databases: J.B., A.D., A.F., C.B., D.W., J.J., M.Ma., R.A., T.Se., W.B., N.T.; provided insights and interpretations: A.D., C.H., C.W., E.J., E.Pa., E.Pr., J.H., J.I., J.J., J.M., K.K., K.N., L.G., M.H., M.Ma., N.T., P.M., R.A., T.L., W.B., W.Y., Y.F., Y.H.; data analysis, tables, figures, and first draft of manuscript: W.B., N.T. All authors edited and approved the final manuscript.

## Competing interests

The authors declare no competing interests.

## Additional information

[1]Ottawa Research and Development Centre, Agriculture and Agri-Food Canada, Ottawa, ON, Canada. [2]Leibniz Institute of Plant Genetics and Crop Plant Research (IPK), Gatersleben, Seeland, Germany. [3]R.W. Holley Center for Agriculture and Health, USDA-ARS, Ithaca, NY, USA. [4]Julius Kuehn Institute, Federal Research Centre for Cultivated Plants, Institute for Breeding Research on Agricultural Crops, OT Groß Luesewitz, Sanitz, MVP, Germany. [5]CSIC, Institute for Sustainable Agriculture (IAS), Córdoba, Spain. [6]Cereal Crops Improvement Research Unit, Edward T. Schafer Agricultural Research Center, USDA-ARS, Fargo, ND, USA. [7]Australian Grain Technologies (AGT), Adelaide, SA, Australia. [8]Crop Improvement and Genetics Research Unit, USDA-ARS, Western Regional Research Center, Albany, CA, USA. [9]School of Integrative Plant Science, Cornell University, Ithaca, NY, USA. [10]Department of Crop Sciences, University of Illinois, Urbana, IL, USA. [11]Crop Development Centre, University of Saskatchewan, Saskatoon, SK, Canada. [12]Agronomy Horticulture and Plant Science Department, South Dakota State University, Brookings, SD, USA. [13]University of California, Davis, Davis, CA, USA. [14]Department of Plant and Agroecosystem Sciences, University of Wisconsin—Madison, Madison, WI, USA. [15]South Australian Research and Development Institute, Department of Primary Industries and Regions, Adelaide, SA, Australia. [16]Louisiana State University, Baton Rouge, LA, USA. [17]Department of Agronomy, National Taiwan University, Taipei, Taiwan, ROC. [18]CBGP, UPM-INIA, Universidad Politécnica de Madrid (UPM), Madrid, Spain. [19]Plant Sciences Department, North Dakota State University, Fargo, ND, USA. [20]Brandon Research and Development Centre, Agriculture and Agri-Food Canada, Brandon, MB, Canada. [21]Department of Plant Science, University of Manitoba, Winnipeg, MB, Canada. [22]Saskatoon Research and Development Centre, Agriculture and Agri-Food Canada, Saskatoon, SK, Canada. [23]Sichuan Agricultural University, Chengdu, China. [24]Department of Agronomy and Plant Genetics, University of Minnesota, Minneapolis, MN, USA. [25]School of Agriculture, Food and Wine, University of Adelaide, Adelaide, SA, Australia. [26]Plant Gene Resources of Canada, Saskatoon Research and Development Centre, Agriculture and Agri-Food Canada, Saskatoon, SK, Canada. [27]Small Grains and Potato Germplasm Research Unit, USDA-ARS, Aberdeen, ID, USA. [28]IBERS, Aberystwyth University, Aberystwyth, UK. [29]Plant and Wildlife Sciences, Brigham Young University, Provo, UT, USA. [30]Institute of Plant Genetics, Breeding and Biotechnology, University of Life Sciences in Lublin, Lublin, Poland. [31]Department of Bioengineering, University of California, Berkeley, CA, USA. [32]German Centre for Integrative Biodiversity Research (iDiv) Halle-Jena-Leipzig, Leipzig, Germany. [33]These authors contributed equally: Raz Avni, Clayton L. Birkett, Asuka Itaya, Charlene P. Wight, Axel Diederichsen, Kathy Esvelt Klos, Yong-Bi Fu, Catherine J. Howarth, Jean-Luc Jannink, Eric N. Jellen, Tim Langdon, Peter J. Maughan, Edyta Paczos-Grzeda, Elena Prats, Taner Z. Sen. ✉e-mail: wubishet.bekele@agr.gc.ca; mascher@ipk-gatersleben.de; nick.tinker@agr.gc.ca

