## [Peer Review file · Nature Communications]

Global genomic population structure of wild and cultivated oat reveals signatures of chromosome rearrangements

Corresponding Author: Dr Wubishet Bekele

Version 0:

Reviewer comments:

Reviewer #1

(Remarks to the Author)

This study collates gbs data from 15 separate studies to produce a kind of genotyping meta study for 9,000 wild and cultivated hexaploid oat accessions. The manuscript is highly successful in its efforts really use genomics to advance the ability of oat researchers to understand their species and ultimately, to accelerate genetic gain in this important and understudied cereal. There are many fascinating insights into the origins of oat species. I found the assignment of *A sativa* to p05 and *A byzantina* to p05 and p06 to be really beautiful. The four broad populations of *A. sterilis* associated with their collection sites and accompanied by major structural variants within these and other population that might well underpin assortative mating and reproductive separation between these populations. There are interesting leads that moves the field on in terms of genomic regions that are associated with adaptation. Phenotypic associations and genetics are beyond the scope of the current work but the work that is presented is clearly a platform for a significant step forward for the oat and wider research community.

I was particularly impressed with the authors efforts to make their datasets FAIR and their analysis really beautifully eg through online PCA plots and interactive maps such as wubi.shinyapps.io/2024-01-11-Avena/.

I found the datasets and their analysis straightforward and really interesting. I am very confident that the work will have high impact and have no suggested changes.

Reviewer #2

(Remarks to the Author)

The manuscript describes an analysis of worldwide diversity and population structure of hexaploid oat, and identifies signatures of structural rearrangements within the germplasm collection. While the study is admirably extensive in terms of accessions analysed, there are some areas of the paper which could be substantially clearer or strengthened.

Introduction:

The authors should make it clearer how this paper relates to the pangenome companion paper. In the conclusions (Line 576), they state that this work helped to “guide the selection of genotypes for sequencing”. This is not made clear in the companion pangenome paper either.

Methods:

- Can the authors please clarify why there is a difference between 9174 accessions in pangenome companion paper compared with 9168 here?
- Have the authors confirmed that all accessions are indeed hexaploid?
- Line 144. “‘Sang’ V1 reference genome, downloaded from the GrainGenes databases.” Please reference the Kamal et al. 2022 paper as well as GrainGenes here.
- Can the authors comment on how successful the use of the Sang reference was? I would expect that given the diversity of the accession panel, some lines would not map so well, and that this could skew the results (such as more missing data for genetically distant taxa). Mapping statistics are required to evaluate this, and should be included in the supplementary files.
- Line 153. The use of different datasets is confusing, and needs to be both justified and clarified. It is not clear when and

why the Matrix50 and 80 were used for the different analyses (are both necessary? If so, why?), so this should be made more explicit in the methods. The additional filtering for the Lostruct analysis is better justified, but adds to the general confusion of data sets used for different analyses.

- Line 173. Please add a reference for the Mahalanobis distance method.

Results:

- Line 228-238. "A phenotypic data analysis was beyond the scope of this study. However, phenotypic data are available for many of the published collections of taxa used in this study (Table 1), which could be used in future studies ...but we draw attention to the population sizes (N) for which observations are available, and 237 to the non-orthogonality of these data, which may preclude many types of statistical analyses". I agree completely that due to the non-orthogonality of quantitative traits, statistical analysis of some of these traits is not appropriate. However, qualitative traits could be used for downstream analysis (such as GWAS), which would significantly increase the value of the paper. Could the authors consider analysing one or two traits which are less likely to be affected by experimental variation (for example, Hull Characteristic has N=857)?

- Line 242 "Missing data". Do you mean missing genotype data?

- Line 262. Could you include an option in shiny to deselect all countries/species...etc? It would be an easier way to only select the country of interest. Also, it would be helpful to colour-code populations in the same way as in the manuscript.

- Line 273. "Mantel test of geographic distances against genetic distances supported a significant correlation ($r=0.1$, $p=0.001$)." This is quite a weak correlation, despite being significant. Also, how many accessions are included in this analysis?

- Line 284-291. This section would benefit from a map as done for *A. sterilis*.

Discussion:

- Line 482-487. "We suggest that the second axis separates the degree of winter habit, while the first axis separates the ability to flower under short daylength..." There is little data presented to support this hypothesis

Conclusion:

- Line 577. "...to guide the selection of genotypes". "Accession" would be more appropriate than "genotype" given how genotype refers to SNP calls elsewhere.

Figures:

- Figure 1.B names in PCA should include non-cultivated species with minor representation. For example, *A. occidentalis* in P04

- Figure 1. Legend. "Species with fewer than four accessions and accessions with ambiguity in classification are omitted". How many?

- Figure 1. Why did you plot only Europe-Middle East?

- Figure S1. Low image quality of the PCA and MDS figures

- Figure 4. Colour scheme (coloured bars) is not colour-blind-friendly, and do not match figure S10 very closely.

- Figure S4 and S5 could be included in shiny

General comments

Are all relevant scripts available somewhere?

Reviewer #3

(Remarks to the Author)

Reviewer #4

(Remarks to the Author)

This manuscript presents a substantial study of hexaploid oat genetic diversity and chromosome rearrangements, analyzing over 9,000 accessions, both wild and cultivated, using the newly available genome sequence. The large sample size, integrated data sources, and online database are significant assets. This work significantly advances our understanding of hexaploid oat diversity and evolution, and provides valuable resources for both oat breeding programs and research into adaptation and domestication. However, the authors should address the following points before publication:

L210-212:

We have concerns regarding the reliability of analyses based on the Matrix50 and Matrix80 datasets. Given the oat genome's large size (approximately 11Gb), the reduction from 115,482 sites in the Matrix50 dataset (9,112 taxa) to only 19,928 sites in Matrix80 (with similar sample size, 8,816 taxa) is substantial. How to ensure that analyses based on this reduced set of loci are reliable and sufficiently representative of the oat genome's diversity? This is particularly crucial considering factors that complicate oat population genetics, such as frequent interspecific gene flow and incomplete lineage sorting. The authors should address how this significant reduction in loci might affect the power and accuracy of downstream analyses, especially in detecting fine-scale population structure or recent evolutionary events.

L242-244:

We understand why the PCA analysis shows a strong bias, due to the high proportion of missing data points. However, the lack of bias in the MDS analysis (Figure S2) does not automatically equate to reliability. Did the authors compare results from analyses using different numbers of loci (e.g., a bootstrap analysis)? A comparison of results across different subsets of loci would help to validate the robustness of the findings.

L253-257:

Figure 1 shows a striking pattern of phylogeographic incongruence within the *A. sterilis* populations (P2, P3, P4 and P5). While geographically distant populations P3 (eastern Mediterranean) and P4 (western Mediterranean) show minimal genetic divergence, the more geographically proximate populations P2 and P5 exhibit greater divergence. This is intriguing, given the discussion (lines 442-460) suggesting a potential link between the *A. sativa* domestication and a specific chromosomal translocation (1A/1C) present in P5 but absent in *A. byzantine*, which probably domesticated from P2. While this translocation is sometimes associated with reproductive isolation, modern hexaploid oat hybrids are generally fertile. The authors need to explain this apparent contradiction between geographic proximity/gene flow and genetic divergence. However, exploration of the evolutionary processes shaping the genetic structure of these *A. sterilis* populations is necessary in the future.

L 284-291:

Thirty-four *A. fatua* accessions falling into nine different populations in the MDS analysis is unexpected. If the MDS results are reliable, phenotypic characterization of these wild samples (including *A. fatua*, *A. hybrida*, and *A. occidentalis*) is essential to confirm their taxonomic assignment. This would help determine if the observed genetic divergence truly reflects distinct lineages or if other factors are influencing the clustering patterns. Further scrutiny of the classification of these accessions is crucial for robust interpretation of the results.

L302,303:

In Figure 2, populations P10 and P21 appear to overlap or cluster closely on the second axis, suggesting they are not effectively separated. Please check.

L328:

Several results in this manuscript rely on data from this citation (Avni R, Author A, Author B, Author C, Author D, Mascher M. Oat Pan Genome Analysis (placeholder for companion paper). *Nature* 999, 1-9 (2024).). We haven't been able to access this cited article. Could the authors provide a download link to the publication?

L545:

Have the authors attempted to incorporate the samples from China in the citation Nan et al. into this manuscript to determine the relationships between those samples and populations P1-P21?

Reviewer #5

(Remarks to the Author)

Version 1:

Reviewer comments:

Reviewer #2

(Remarks to the Author)

Thank you for carefully addressing my comments in the revised manuscript. I have only one small comment/query on the alignment statistics which were helpfully added to Supplementary Data S1; some of the PF_%Aligned values for samples included in the Matrix50 data set are very low - I realise this is a very small proportion of lines, but can these samples be considered reliable?

(Remarks on code availability)

Some example code is provided, but the file is not all readable or commented.

Reviewer #3

(Remarks to the Author)

(Remarks on code availability)

Reviewer #4

(Remarks to the Author)

Thanks for your detailed responses. I have no further comments.

(Remarks on code availability)

Reviewer #5

(Remarks to the Author)

(Remarks on code availability)

Version 2:

Reviewer comments:

Reviewer #2

(Remarks to the Author)

Thank you for your detailed responses. I have no further comments.

(Remarks on code availability)

Code is now clearer, but I have not tried running it.

Reviewer #3

(Remarks to the Author)

(Remarks on code availability)

Dear referees,

Thank you for reviewing our manuscript. Here is a point-by-point response to all your comments.

RESPONSE TO REVIEWER COMMENTS**Reviewer #1 (Remarks to the Author):**

This study collates gbs data from 15 separate studies to produce a kind of genotyping meta study for 9,000 wild and cultivated hexaploid oat accessions. The manuscript is highly successful in its efforts really use genomics to advance the ability of oat researchers to understand their species and ultimately, to accelerate genetic gain in this important and understudied cereal. There are many fascinating insights into the origins of oat species. I found the assignment of *A sativa* to p05 and *A byzantina* to p05 and p06 to be really beautiful. The four broad populations of *A. sterilis* associated with their collection sites and accompanied by major structural variants within these and other population that might well underpin assortative mating and reproductive separation between these populations. There are interesting leads that moves the field on in terms of genomic regions that are associated with adaptation. Phenotypic associations and genetics are beyond the scope of the current work but the work that is presented is clearly a platform for a significant step forward for the oat and wider research community.

I was particularly impressed with the authors efforts to make their datasets FAIR and their analysis really beautifully e.g. through online PCA plots and interactive maps such as wubi.shinyapps.io/2024-01-11-Avena/.

I found the datasets and their analysis straightforward and really interesting. I am very confident that the work will have high impact and have no suggested changes.

A: Thank you very much for these observations and the encouraging review.

Reviewer #2 (Remarks to the Author):

The manuscript describes an analysis of worldwide diversity and population structure of hexaploid oat, and identifies signatures of structural rearrangements within the germplasm collection. While the study is admirably extensive in terms of accessions analysed, there are some areas of the paper which could be substantially clearer or strengthened.

Introduction:

The authors should make it clearer how this paper relates to the pangenome companion paper. In the conclusions (Line 576), they state that this work helped to "guide the selection of genotypes for sequencing". This is not made clear in the companion pangenome paper either.

A: This work was initiated simultaneously with the pan genome effort, in part to guide the selection of genotypes for sequencing. We have revised and highlighted this interdependence in both papers as follows:

- We clarified the interdependence by including this in the first objective: “(i) to assemble and analyze a GBS data set that captures the genetic diversity in a large and globally representative set of wild and cultivated hexaploid oat and guides selection of a representative pan genome”.
- A discussion of how the reference sequences from the oat pan genome fit into our diversity space is included in the new Supplementary Analysis S3.
- For example, we commented in both manuscripts that there is no sequenced accession from population P2. This was because P2 was not represented in our first data set and only became evident after we genotyped additional *A. sterilis* lines from the BioMob project (Table 1).

Methods:

- Can the authors please clarify why there is a difference between 9174 accessions in pangenome companion paper compared with 9168 here?

A: These are the initial numbers prior to filtering. This difference was due to removal of some non-hexaploid lines in the current work. We have removed some additional taxa so the total of unfiltered taxa in the metadata is now 9153, but the filtered lines remain the same. Numbers will be corrected in the other manuscript when submitted.

We also highlighted (in both papers) that we used a different reference genome to conduct the SNP calls. Here we used the short-read assembly of cv. ‘Sang’ as the reference, since the PanOat genomes were not originally available. In Avni et al we used the long-read assembly of cv. GS7 as the reference because of its superior contiguity and also to confirm the robustness of the analysis. A comparison of both SNP calls was added to Supplementary Analysis S2. We conclude that there is very little difference between these different SNP calls at the level of analysis and interpretation performed in this study.

- Have the authors confirmed that all accessions are indeed hexaploid?

A: We have not cytologically confirmed the hexaploidy of all the samples. The majority of samples are from cultivated *A. sativa*, which would not be confused with any non-hexaploid species due to its unique adaptation to cultivation, which is well-known to the contributing oat breeders. Samples from Yan et al., 2016 (Table 1) were analyzed previously by flow cytometry, and a small number of corrections were made, as described in that study. Some additional samples of questionable phenotype have been karyotyped by one of the authors (Rick Jellen) and corrected or removed as needed.

As a precaution, we also examined the full data set to confirm that all samples contained genotype calls on all chromosomes, including the D-genome chromosomes (the D genome is not present in any known diploids or tetraploids). We have added a statement in the second

paragraph of the results to emphasize this: “All accessions contained SNPs on all 21 chromosomes, confirming the hexaploid nature of all samples.” (line 139).

- Line 144. "Sang' V1 reference genome, downloaded from the GrainGenes databases." Please reference the Kamal et al. 2022 paper as well as GrainGenes here.

A: Done. (We removed GrainGenes as a reference and replaced it with the Kamal et al 2022 reference genome paper, since there are multiple places where the genome can be downloaded).

- Can the authors comment on how successful the use of the Sang reference was? I would expect that given the diversity of the accession panel, some lines would not map so well, and that this could skew the results (such as more missing data for genetically distant taxa). Mapping statistics are required to evaluate this and should be included in the supplementary files.

A: We selected the Sang genome as a reference because it was the best-annotated genome at the project's onset. However, we repeated the analysis in the PanOat paper (Avni et al.) using a different reference (GS7; see above) which provided a very similar MDS analysis. Furthermore, we conducted the entire analysis using alternative reference-free SNP calls, also giving very similar results. We now provide Supplementary Analysis S2 to compare these different SNP calls. This includes a correlation analysis between genetic distances calculated with different SNP calls showing ($r=0.9935$) with a significance of $P<0.001$ based on 999 permutations.

As requested, we also computed the alignment statistics which are now included in the Metadata (Supplementary Data S1). Although there was a huge variation in number of reads and rate of alignments, the average number of reads per accession (3.45 million) is higher than previous similar studies and gave a comparable average proportion of aligned reads (0.97).

As another precaution, we analysed whether the proportion of called SNPs was different among the major species, possibly due to ascertainment bias caused by alignment differences to an *A. sativa* reference. Our analysis (presented in Supplementary Table S2 in Supplementary Analysis S2) shows that the use of an *A. sativa* reference genome did not affect the proportion of SNPs called in taxa from *A. byzantina* or *A. sterilis*. In fact, if we look at lines with similar read depth, we achieved a higher proportion of complete genotypes in *A. sterilis* than in *A. sativa*.

We also had concerns that chromosome rearrangements could affect the analysis, since SNPs would be assigned to the chromosome configuration of the reference that was used. However, the presence of translocations or inversions in a reference genome still allows SNPs to be called. Moreover, the resulting LD caused by SNPs that have “moved,” but are called relative to a single chromosome configuration, is probably the main reason that we can detect chromosome rearrangements based on this population data. The presence of large deletions in the reference genomes would be of more concern, but the pangenome reveals this is not a factor.

- Line 153. The use of different datasets is confusing and needs to be both justified and clarified. It is not clear when and why the Matrix50 and 80 were used for the different analyses (are both

necessary? If so, why?), so this should be made more explicit in the methods. The additional filtering for the Lostruct analysis is better justified but adds to the general confusion of data sets used for different analyses.

A: Matrix80 was used for analyses requiring a more complete data set. Matrix50 provides the community with access to more markers, and to taxa with sparse markers that were dropped from Matrix80, which may be useful in other future analyses. Matrix80 was required for the sNMF method, while Matrix50 was used to place the full set of taxa on the MDS plot. We have now clarified this in the methods section with an additional statement: “Matrix50 provides the community with a larger set of markers and taxa, while Matrix80 was used for analyses that were more sensitive to missing data, as described below”. (line 514).

We have also clarified this in Supplementary Analysis S2. In the online R-Shiny resources, lines that are not included in Matrix 80 (but are still in Matrix 50) are shown in grey (not assigned) when the “Populations” view is selected. We have found that users often look for a “favorite variety” that was dropped from Matrix 80, and this helps to explain to them what happened.

- Line 173. Please add a reference for the distance method.

A: Unfortunately, it was necessary to reduce the number of references to comply with journal requirements. The proper Mahalanobis citation is from 1936 and is no longer accessible except in re-published formats. In this case we feel the method is adequately described and credited in the reference that is given for the PCAdapt package. We reworded the reference slightly to make this clear: “we applied the Mahalanobis distance method within the PCAdapt package.” (Line 531)

Results:

- Line 228-238. "A phenotypic data analysis was beyond the scope of this study. However, phenotypic data are available for many of the published collections of taxa used in this study (Table 1), which could be used in future studies ...but we draw attention to the population sizes (N) for which observations are available, and 237 to the non-orthogonality of these data, which may preclude many types of statistical analyses". I agree completely that due to the non-orthogonality of quantitative traits, statistical analysis of some of these traits is not appropriate. However, qualitative traits could be used for downstream analysis (such as GWAS), which would significantly increase the value of the paper. Could the authors consider analysing one or two traits which are less likely to be affected by experimental variation (for example, Hull Characteristic has N=857)?

A: Thank you. There are indeed some potentially informative contrasts that can be evaluated using the classification data. We opted to provide this information in semi-analyzed format (i.e. means with heatmap, counts, and standard deviations) that allows simple visual inspection of potential differences among populations, and simple T tests or multiple contrasts if desired. We hope that this will inspire further hypotheses and experiments within the oat community. However, due to space limitations, and the speculative nature of these simple unplanned comparisons, we opted to provide only the examples that were mentioned, and to suggest these as starting points for further work.

The companion manuscript, and multiple published datasets included in our study showed the utility of these datasets in identifying kmers/SNPs and large-scale chromosomal inversion for heading association among cultivated lines from the CORE, and other experiments.

- Line 242 "Missing data". Do you mean missing genotype data?

A: Yes, thanks. We revised this to "missing genotype data."

- Line 262. Could you include an option in shiny to deselect all countries/species...etc? It would be an easier way to only select the country of interest. Also, it would be helpful to colour-code populations in the same way as in the manuscript.

A: Thank you for this suggestion. There is a feature to isolate one group by double-clicking on the item in the legend. Unfortunately, the points disappear completely when they are deselected such that one loses the context of the MDS. For this reason, we added an option for the user to select or modify colors. We felt this solution was more flexible, especially for users with diverse colour perceptions. We combined this option with colour schemes that were slightly monochromatic (or pale pastels in the case of countries) such that users can easily select a contrasting colour to highlight their population, species, or project of interest. As an example, to highlight "Turkey" as a country of origin, go to the bottom left, select Turkey, and choose a unique and contrasting colour (e.g. bright orange).

By default, the populations are actually coloured in the same way as the figures in the manuscript, however this is only true for the set of K=21 populations. For K=12 or K=16, the populations are coloured in arbitrary colour selections. These different K values are provided for comparison to K=21. We have now made "K=21" into the default option for populations.

We appreciate that these "tricks" in the R-Shiny app may not be obvious or intuitive. Therefore, we have created a simple tutorial to go with the shiny app. The tutorial is stored on the GrainGenes project page and can also be accessed by link in the "about" page of the Shiny apps.

- Line 273. "Mantel test of geographic distances against genetic distances supported a significant correlation ($r=0.1$, $p=0.001$)." This is quite a weak correlation, despite being significant. Also, how many accessions are included in this analysis?

A: This test includes all wild and landrace accessions for which collection sites were available (N=2279). We have added this information to the caption for Figure S6. We recognize that the correlation is not strong, but there is an important reason for this (see also comment and reply "L253-257" from Reviewer 4). To reflect this, we have made the following revision at line 204: "..... supported a weak-but-significant correlation ($r=0.1$, $p=0.001$), while the scatter plot of these distances illustrated some tendency for smaller genetic distances to be found among the most proximally collected accessions (Figure S6). The reason that this correlation is not stronger probably relates to the fact that some geographically distant populations of *A. sterilis* (e.g. P03 vs. P04) share more genetic similarity than some proximal populations (e.g. P02 vs. P05)."

- Line 284-291. This section would benefit from a map as done for *A. sterilis*.

A: The map-based analysis was performed for the non-cultivated (collected) non-sterilis lines referred to in this paragraph. The analysis is presented in Figure S7 (the map is S7B).

Unfortunately, a map-based analysis would not be accurate or useful for the analysis of cultivated varieties. While we can identify the original breeder of most varieties, many breeding programs develop varieties for broad areas of adaptation. Furthermore, all varieties from a given program would be placed on top of each other at the breeder's home location.

Discussion:

- Line 482-487. "We suggest that the second axis separates the degree of winter habit, while the first axis separates the ability to flower under short daylength..." There is little data presented to support this hypothesis.

A: This was a descriptive observation, but the underlying cause was speculative. We have now omitted this paragraph since the discussion was already very long and adding further caveats or additional analyses here would be difficult.

Conclusion:

- Line 577. "...to guide the selection of genotypes". "Accession" would be more appropriate than "genotype" given how genotype refers to SNP calls elsewhere.

A: Thank you, we have made this replacement.

Figures:

- Figure 1.B names in PCA should include non-cultivated species with minor representation. For example, *A. occidentalis* in P04

A: Listing the minor species here would clutter the figure and detract from the focus on *A. sterilis*. Many groups contain a small number of *A. fatua* lines, and the *A. occidentalis* classification is contested by some taxonomists. We deal with these minor species later. To emphasize this and correct the figure we added the word "primarily" in the caption: "Four populations composed primarily of *Avena sterilis* (P02, P03, P04, P05) are indicated".

- Figure 1. Legend. "Species with fewer than four accessions and accessions with ambiguity in classification are omitted". How many?

A: This has now been simplified to read "A. hausknechtii, a contested species with a single duplicated accession in P04 is omitted" (Line 652). We have now included *A. ludoviciana* in these counts because we revised two *A. sterilis* classifications based on corrections from the contributor. We also draw attention to an expanded discussion of species classifications in Supplementary Note S1 where *A. ludoviciana* is mentioned.

- Figure 1. Why did you plot only Europe-Middle East?

A: We assume that you mean the *A. sterilis* accessions that were cut off from Figure 1b (if you are referring to other species like *A sativa*, please see response to Line 284-291 above). In Figure 1b, accessions were cut off from Southern Europe and the Horn of Africa. We have now expanded the figure to include these accessions. There is still one accession of *A. occidentalis* from Saskatchewan Canada that is cut off from the figure (but visible in the online map browser). We think this is an anomaly: there is a working hypothesis in the community that *occidentalis* is a fatua-like mutation from *A. sterilis* that should not be found in Canada unless it escapes as a weed. However, this is not a question we can answer in this paper.

- Figure S1. Low image quality of the PCA and MDS figures

A: Supplementary Figure S1 has now been improved and expanded. The previous plots appeared fuzzy because there was a very small font for the axes, and the axes were inverted to show the plots on the same orientation. In the latest version of Figure S1 (which now contains 6 plots) we have removed the axis labels, since the axes scales are arbitrary in the MDS analysis, and since we are only interested in the shape of the PCA plots and not the scale.

- Figure 4. Colour scheme (coloured bars) is not colour-blind-friendly, and do not match figure S10 very closely.

A: We have replaced the colour bars in Figure 4 with the more colour-blind-friendly colours in the related Figure S10.

- Figure S4 and S5 could be included in shiny

A: We do not see an obvious way to include the admix data in the existing MDS based Shiny app, so this would require the development of a separate app specifically to display the admixture plots. Since the population structure information is captured by the Shiny app, we do not feel this would be meaningful or useful for most readers. In the revised manuscript, all data from all figures are included in a source data file, as requested by the journal. This would allow readers with a specific interest to further investigate the admix or other plots/datasets.

General comments

Are all relevant scripts available somewhere?

A: We have now added a “code availability section” which points to a supplemental archive (Supplementary Data S10) which provides and annotates all reusable code that we developed (i.e. beyond simple command-line execution of programs and third-party code which is referenced in the narrative).

Reviewer #3 (Remarks to the Author):

I co-reviewed this manuscript with one of the reviewers who provided the listed reports. This is

part of the Nature Communications initiative to facilitate training in peer review and to provide appropriate recognition for Early Career Researchers who co-review manuscripts.

A: Thank you.

Reviewer #4 (Remarks to the Author):

This manuscript presents a substantial study of hexaploid oat genetic diversity and chromosome rearrangements, analyzing over 9,000 accessions, both wild and cultivated, using the newly available genome sequence. The large sample size, integrated data sources, and online database are significant assets. This work significantly advances our understanding of hexaploid oat diversity and evolution and provides valuable resources for both oat breeding programs and research into adaptation and domestication. However, the authors should address the following points before publication:

L210-212:

We have concerns regarding the reliability of analyses based on the Matrix50 and Matrix80 datasets. Given the oat genome's large size (approximately 11Gb), the reduction from 115,482 sites in the Matrix50 dataset (9,112 taxa) to only 19,928 sites in Matrix80 (with similar sample size, 8,816 taxa) is substantial. How to ensure that analyses based on this reduced set of loci are reliable and sufficiently representative of the oat genome's diversity? This is particularly crucial considering factors that complicate oat population genetics, such as frequent interspecific gene flow and incomplete lineage sorting. The authors should address how this significant reduction in loci might affect the power and accuracy of downstream analyses, especially in detecting fine-scale population structure or recent evolutionary events.

A: It is correct that most analyses that we conducted were based on the Matrix80 data set with the reduced number of loci. The larger Matrix50 is used only for the descriptive MDS analyses, but it is also available to readers who may wish to filter it differently (e.g. fewer taxa and more loci) for other purposes (e.g. GWAS analysis among a more restricted taxa set). Despite the reduction from 115K sites to about 20K SNP sites, the smaller data set contains adequate genome coverage for most types of analyses. We did have some concern that gaps or regions with low SNP coverage may affect the estimates in the extent of structural anomalies. We have now brought attention to this concern in the manuscript (line 269): ("We note that these results may also be affected by marker density and/or polymorphism content. For example, low marker density on chromosome 7D may be the reason that the detected outlier window (256.6Mbp - 412.1Mbp) is smaller than the size of the physical inversion").

We have further addressed the concern about data robustness in the next comment.

L242-244:

We understand why the PCA analysis shows a strong bias, due to the high proportion of missing data points. However, the lack of bias in the MDS analysis (Figure S2) does not automatically equate to reliability. Did the authors compare results from analyses using different numbers of loci (e.g., a bootstrap analysis)? A comparison of results across different subsets of loci would

help to validate the robustness of the findings.

A: We realize that the use of the larger and sparser (Matrix50) data set for MDS analysis raises concerns and has caused some confusion. We included it so that (A) readers can evaluate alternate data filtering methods if they wish and (B) there were at least some descriptive results for lines with low marker coverage. The MDS plot with sparse data is intended to be descriptive, complementing the structure analysis in the more complete data from which most conclusions are drawn. We have added a statement (Lines 164-168) to help clarify this, and we have expanded the supplementary analysis (Supplementary Discussion S2) to address concerns about robustness and repeatability further. The new supplementary analysis compares PCA and MDS, showing that they are almost identical in Matrix80 and that they are nearly identical when analyzed by a different reference genome or by the alternate reference-free (Haplotag) data. We also show that there is a very high ($r=0.9935$) Pearson correlation coefficient between genetic distances based on SNPS that are called independently by different methods, significant at $P<0.001$ based on 999 permutations.

L253-257:

Figure 1 shows a striking pattern of phylogeographic incongruence within the *A. sterilis* populations (P2, P3, P4 and P5). While geographically distant populations P3 (eastern Mediterranean) and P4 (western Mediterranean) show minimal genetic divergence, the more geographically proximate populations P2 and P5 exhibit greater divergence. This is intriguing, given the discussion (lines 442-460) suggesting a potential link between the *A. sativa* domestication and a specific chromosomal translocation (1A/1C) present in P5 but absent in *A. byzantine*, which probably domesticated from P2. While this translocation is sometimes associated with reproductive isolation, modern hexaploid oat hybrids are generally fertile. The authors need to explain this apparent contradiction between geographic proximity/gene flow and genetic divergence. However, exploration of the evolutionary processes shaping the genetic structure of these *A. sterilis* populations is necessary in the future.

A: Thank you for suggesting further clarification of this interesting observation. The presence of different wild populations in the same area (and similar populations in different areas) seems unusual if all hexaploids are inter-fertile. At line 204, we have now added some discussion as a new paragraph in relation to another reviewer's comment (see Reviewer 2, line 273, re Mantel test). We agree with this reviewer that the resolution of this phenomenon may require further experimentation, and we have also noted this in this new paragraph.

L 284-291:

Thirty-four *A. fatua* accessions falling into nine different populations in the MDS analysis is unexpected. If the MDS results are reliable, phenotypic characterization of these wild samples (including *A. fatua*, *A. hybrida*, and *A. occidentalis*) is essential to confirm their taxonomic assignment. This would help determine if the observed genetic divergence truly reflects distinct lineages or if other factors are influencing the clustering patterns. Further scrutiny of the classification of these accessions is crucial for robust interpretation of the results.

A: We thank the reviewers for raising this topic, which is of interest to many members of the oat research community. The fatuoid (seed shattering) character that defines these species is easy to score and unlikely to have been a classification error in so many accessions. In fact, some authors did expect to see *A. fatua* falling into multiple populations, since there is a proposal among some members of the community that these three species defined by the fatuoid character should never have been considered as separate species; that they are natural single-gene mutations that have arisen multiple times in both wild and cultivated oat populations.

Here, as elsewhere, we have avoided presenting opinions on taxonomy. Most authors do not consider themselves to be taxonomists, and while our data may support taxonomic inferences, we defer these inferences to focused studies by this community of experts.

To help orient the reader to this, and to raise the possibility of a species reclassification, we have now added a small amount of background to this observation: *“The fatuoid (floreit shattering) character that defines these three species has long been understood as a single recessive genetic factor (BAUM 1977) however, it has not been resolved whether this character arises through mutation, chromosome aberration, or hybridization. Our observation of A. fatua accessions within multiple populations of cultivated oat suggests that many of these lines have a recent and spontaneous origin. Once established, these fatuoid shattering lines could find local or temporary selective advantage within cultivated crops.” (lines 218-223)*

L302,303:

In Figure 2, populations P10 and P21 appear to overlap or cluster closely on the second axis, suggesting they are not effectively separated. Please check.

A: We are sorry, our wording here was not clear. We meant that “the second axis separates P10 and P21 from most other populations”. We have revised this to clarify on line 233.

L328:

Several results in this manuscript rely on data from this citation (Avni R, Author A, Author B, Author C, Author D, Mascher M. Oat Pan Genome Analysis (placeholder for companion paper). Nature 999, 1-9 (2024).). We haven't been able to access this cited article. Could the authors provide a download link to the publication?

A: This companion manuscript was provided in our submission and should have been available to reviewers. Unfortunately, this may have been hidden or neglected. At this point, the Avni et al. manuscript is under a second review with the journal Nature, while a preprint is now available here: <https://www.biorxiv.org/content/10.1101/2024.10.23.619697v1>.

L545:

Have the authors attempted to incorporate the samples from China in the citation Nan et al. into this manuscript to determine the relationships between those samples and populations P1-P21?

A: Unfortunately, it is not feasible to incorporate the data from the Nan study, since it was performed using whole-genome resequencing rather than GBS. The samples themselves would also not be available to us due to restrictions on sharing Chinese germplasm. We note that our previous work (Yan et al., 2020) contributed 190 Chinese naked oat lines (mostly landraces) to our current study. Of these, 39 landraces were in common with Nan et al. However, the Nan et al study included only 22 covered oat lines, of which four were from North America (Buck, Ajax, Nuprime, and Stout). Not only is this a very small sample to make a statement on covered oat diversity, it also does not represent North America, where much of the cultivated oat diversity resides. Thus, we suspect that important components of diversity are lost in their analysis due to sample size. As an example, the Nan study also contains only a single *A. byzantina* line and a single *A. sterilis* line, yet these two lines are not separated at all from other lines on their PCA (separation by factorial analysis would require an adequate number of samples such that the diversity is captured in the first two components). The same thing may have happened to their covered samples.

Reviewer #5 (Remarks to the Author):

A: Thank You!

Dear referees,

Thank you for reviewing our manuscript. Here is a point-by-point response to your comments.

RESPONSE TO REVIEWER COMMENTS**Reviewer #2 (Remarks to the Author):**

Thank you for carefully addressing my comments in the revised manuscript.

I have only one small comment/query on the alignment statistics which were helpfully added to Supplementary Data S1; some of the PF_%Aligned values for samples included in the Matrix50 data set are very low - I realise this is a very small proportion of lines, but can these samples be considered reliable?

A: Thank you for this question. Our filtering in this project was based on completeness of the SNP calls rather than on the number or proportion of aligned reads. However your suggestion has helped us to clarify this, and to address whether samples with low numbers of aligned reads are reliable. To further address this comment, we have added five additional fields to the metadata (as described in the field headings in Sheet1 of DataFile_S1_Metadata):

Matrix50: These taxa were included in Matrix50 (i.e. they are placed on the MDS plots but not used in the structure analyses).

PMissTaxa50: Proportion of missing genotypes for this taxa in Matrix50.

Matrix80: These taxa were included in Matrix80 (i.e. used for population structure analysis).

PMissTaxa80: Proportion of missing genotypes for this taxa in Matrix80. Most of the taxa have complete data through the imputation step. Those with very low read counts have localized gaps that cannot be imputed.

Lostruct: These taxa were included in the Lostruct analysis. Those with more than 50% missing have been dropped from Matrix80 for this analysis.

We have also dropped the proportion of aligned reads, keeping only the total number of aligned reads. The proportion can be computed from the total read count, and is less important than the total aligned reads.

We have previously addressed comments regarding the reliability of Matrix50: i.e. we consider it informative mainly for the approximate placement of taxa in the MDS plot, but the data were not used for population inference or structure analyses. Taxa in Matrix80 were used for the population structure analyses, which was used to infer population membership. The matrix was further trimmed for the Lostruct analysis, as described in the methods. The new metadata now shows clearly which taxa were used in each analysis.

It can now be seen that the minimum number of aligned reads for inclusion in Matrix80 was 150127, while the minimum for inclusion in Lostruct analyses was 256581. These lower read counts are not

ideal, but they appear to support a robust analysis as shown by the summary in Supplementary Discussion S2.

We have clarified this in the revised manuscript (Line 152), drawing attention to the additional metadata, and advising caution regarding the interpretation of individual taxa with low read counts.

Reviewer #2 (Remarks on code availability):

Some example code is provided, but the file is not all readable or commented.

A: Thank you for this comment. While we do not consider the provided code to be central to this manuscript, we have now improved the description of the R scripts and the small Pascal program that we used, and have now included readme files and sample input for the Pascal program. This is described in an updated “Code Availability” section (Line 560), as well as within the self-contained archive “DataFile_S10_Custom_Code.gz”. We have been actively engaged in transferring these methods to students and members of the oat community, and this improved annotation will be appreciated.

Reviewer #3 (Remarks to the Author):

Thank you.

Reviewer #4 (Remarks to the Author):

Thanks for your detailed responses. I have no further comments.

Thank you.